# Monitoring of species' genetic diversity in Europe varies greatly and overlooks potential climate change impacts

Genetic monitoring of populations currently attracts interest in the context of the Convention on Biological Diversity but needs long-term planning and investments. However, genetic diversity has been largely neglected in biodiversity monitoring, and when addressed, it is treated separately, detached from other conservation issues, such as habitat alteration due to climate change. We report an accounting of efforts to monitor population genetic diversity in Europe (genetic monitoring effort, GME), the evaluation of which can help guide future capacity building and collaboration towards areas most in need of expanded monitoring. Overlaying GME with areas where the ranges of selected species of conservation interest approach current and future climate niche limits helps identify whether GME coincides with anticipated climate change effects on biodiversity. Our analysis suggests that country area, financial resources and conservation policy influence GME, high values of which only partially match species' joint patterns of limits to suitable climatic conditions. Populations at trailing climatic niche margins probably hold genetic diversity that is important for adaptation to changing climate. Our results illuminate the need in Europe for expanded investment in genetic monitoring across climate gradients occupied by focal species, a need arguably greatest in southeastern European countries. This need could be met in part by expanding the European Union's Birds and Habitats Directives to fully address the conservation and monitoring of genetic diversity.

The maintenance of wild population genetic diversity (PGD) is an important component of the Convention on Biological Diversity (CBD)[1], but it has received little international attention until recently[1–4], limiting our ability to monitor and manage wild populations to sustain PGD[5]. The resulting urgent need for expanded monitoring of PGD motivates the development of globally implementable indicators of genetic diversity[6–9], some of which are included in the recently adopted CBD Kunming-Montreal Global Biodiversity Framework[3,10]. But while ongoing anthropogenic loss of PGD is being documented[11–13], efforts to detect climate change effects on PGD are taxonomically and geographically limited[14,15] and are absent from international biodiversity agreements. Populations in extreme climatic conditions, such as those near trailing climatic niche margins, are particularly relevant to species' potential for adaptation to a changing climate[16]. Nonetheless, multispecies patterns of populations near trailing niche margins, which can serve as potential indicators of areas important for the adaptive potential of multiple species and thus reveal possible PGD monitoring sites, remain unidentified. This suggests the need for improved quantification of the relationships between species' niche limits along environmental gradients and associated PGD[17,18].

✉e-mail: peter.b.pearman@gmail.com

Species populations close to their environmental niche margins may differ genetically from those at the niche centre and influence the course of adaptation to changing environments[19,20]. Evidence shows that populations at niche margins towards stressful environmental extremes are locally adapted[21], having distinguishable genetic architecture independent of their geographic position within the species range[22]. Populations near trailing niche limits probably hold important, adaptive genetic variants[22–24] that can reduce predicted range loss[18,25] and contribute to the adaptation of environmentally central populations[26] to a warming, drying climate, despite greater gene flow from the niche centre to these marginal populations[27]. But genetic diversity and adaptive variants held in marginal populations may be lost (1) when gene flow to environmentally central areas is impeded, (2) when genetic drift strongly affects populations with small effective population sizes or (3) if the populations go extinct as climate extremes eventually exceed species' tolerances[28]. These results suggest that global genetic monitoring frameworks[10] need to anticipate climate impacts, collect samples across entire climate gradients and evaluate the contributions of marginal populations to genetic diversity and adaptive potential[29]. However, no previous accounting of recent and historical PGD monitoring exists, leaving us ignorant of taxonomic, national and geographic trends in monitoring effort, and hampering our capacity to detect changing PGD and adaptive potential under climate change threat. Yet, even without such accounting, existing PGD monitoring efforts suggest notable resources, infrastructure and political support, and can serve as an index of current and potential future genetic monitoring effort (GME).

Here we examine the gap between GME and the need for genetic monitoring generated by deteriorating climatic conditions by asking the following questions. (1) How is GME distributed across Europe, and on which taxa has PGD monitoring focused? (2) Which factors explain among-country variation in GME? (3) How will countries differ in the exposure of threatened species to climate change? Finally, (4) how does GME coincide with anticipated impacts of climate change on habitat suitability for populations? Using evidence of monitoring from the peer-reviewed and technical literature, we examine how 38 countries in the European Commission's Cooperation in Science and Technology (COST) programme[30] demonstrate GME for purposes of biodiversity conservation and management. The collective use of COST full-member countries as a study area allowed us to cover much of the European continent and major islands. We explain variation in GME in these countries in relation to two fundamental national characteristics: per capita gross domestic product (GDP) and area. One could also expect greater GME in southern Europe in recognition of greater habitat diversity, species endemicity and biodiversity hotspots than in the north[31,32].

We used climate and biological data to stratify species ranges into areas with core climatic niche conditions and areas with conditions near niche limits (that is, areas of niche marginality), distinguishing areas with trailing niche margins due to climate change. We then compared a multispecies indicator of trailing niche marginality to country GME, to directly relate GME to climate-driven decline in niche conditions for multiple species. To do this, we estimated and mapped the range-wide predicted impacts of climate change on the present and future geographic distributions of climatic niche marginality[33]. We did this for species in four groups, selected for recognized and potential conservation and management interest (amphibians, large birds, carnivorans and forest trees). Within COST member countries, we aggregated the climate change impacts on these groups of species by tallying a count of niche marginal species, thereby defining the pattern of trailing climate niche marginality among countries. Finally, we plotted this indicator of cumulative climate impacts on species against values of country GME.

## Results

Between 22 November 2019 and 31 December 2021, we received 480 submissions of candidate monitoring projects from conservation

**Table 1 | Requested information to characterize submitted monitoring projects/programmes**

| Variable | Values |
|---|---|
| Contributor | First and last name(s) |
| Description of project | Text description provided by contributor |
| Programme/project name | Text name, not available |
| Barcoding study | True/false |
| Within-species diversity | True/false |
| Temporal category | 'Snapshot', 'Horizontal' |
| Annual sampling? | True/false |
| Country | One or more country names |
| Political extent | Regional, national, multi-national |
| Marker type | Organelle sequence, other autosomal, SNP, microsatellite, sex chromosome, multi-marker |
| Strict/relaxed | 'Strict' indicates study was a priori designed as a monitoring study; 'relaxed' if data used post-hoc for monitoring |
| Focal groups (true/false) | Carnivora, bear, wolf, lynx, other mammal, Aves, Insecta, fish, marine, plant, forest trees, amphibians, other, domesticated/captive |
| Name(s) of focal taxon/taxa | Common names (English), scientific names |
| EU Directive (Habitats or Birds) and Annex | EU Directive and Annex listing for each monitored species |
| Documentation/document type | Project report in national language, project report in English, government report in national language, other report in national language, scientific publication, not available |
| Document format | PDF, link, paper copy, not available |
| Document locator | DOI if available |
| Document title or reference | Complete citation when available |
| Project or report webpage | URL listed when available |
| Notes | Unrestricted text |

geneticists, practitioners and stakeholders. These submissions responded to a variety of data fields that described candidate projects (Table 1). We evaluated these for validity as Category II genetic monitoring projects[34], which report temporally separate assessments of PGD metrics of one or more populations of a species. We focus here exclusively on this type of genetic monitoring because it directly tracks PGD over time, while we recognize that other types of genetic monitoring, including genetic assessments and species identification programmes, are also highly relevant to conservation but address questions other than the change in PGD over time. We found 38 additional candidate Category II monitoring projects through a structured search of the Web of Science. Of the total 518 candidates, we identified 103 as valid Category II monitoring projects, the vast majority of which report sampled populations from one (84) or two (14) countries[35]. We tallied international and transboundary projects separately by country, and we documented a total of 151 national-level projects of Category II genetic monitoring. We found Category II monitoring in 30 of 38 COST countries that were full members at the beginning of data solicitation (Extended Data Fig. 1a,b).

### GME
We documented a maximum of 12 projects for Belgium and Sweden and 11 projects for Spain and France (Extended Data Fig. 1a). We found no GME in eight countries (Extended Data Fig. 1b), including ones as

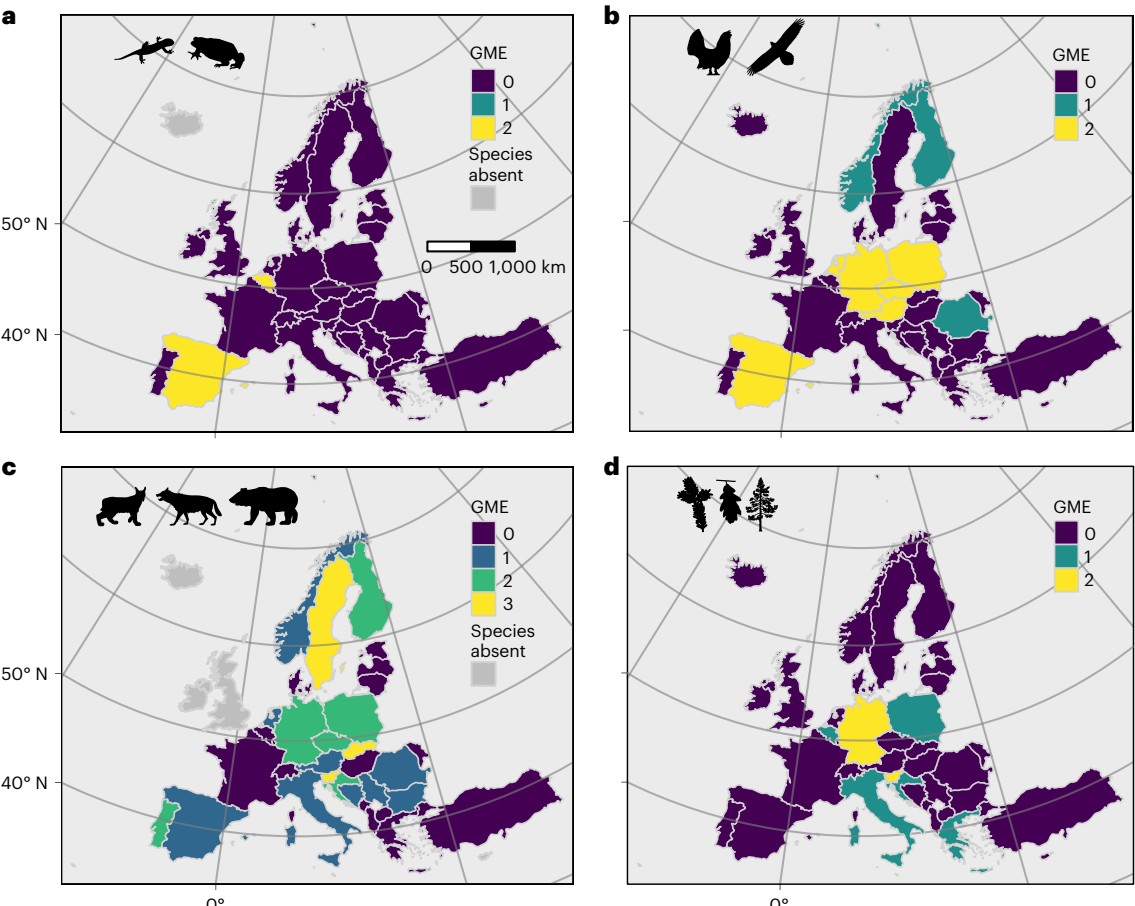

**Fig. 1 | Geographic distribution of effort to monitor population genetic diversity (GME), for purposes of conservation or management, among COST full-member countries. a–d**, The tally of genetic monitoring programmes for amphibians (**a**), birds (**b**), carnivorans (**c**) and forest trees (**d**). The programmes included here are consistent with the requirements for Category II monitoring, and they offer documentation of multiple estimates over time of at least one index of genetic diversity. Few countries have GME for amphibians, while most countries have established at least one programme for a carnivoran species.

geographically and economically disparate as Turkey and Luxemburg. This pattern is robust to the exclusive consideration of terrestrial wild species (that is, the exclusion of programmes monitoring fish, marine species and domesticated/captive populations; Extended Data Fig. 2). The GME of COST countries varies greatly by taxonomic and functional groups. For example, while many amphibians are of recognized conservation concern, only two European countries demonstrated GME for amphibians (Belgium and Spain; Fig. 1a). Many more countries (nine) have monitored PGD in at least one bird species (Fig. 1b and Extended Data Fig. 3b). Approximately half of COST countries (18) have monitored PGD in one or more large carnivorans (Fig. 1c and Extended Data Fig. 3c), although certain carnivorans are absent from some COST countries (Extended Data Fig. 4a,c,e). In contrast, while all COST countries have tree species, less than one quarter of COST countries (seven) have monitored PGD in at least one of these species (Fig. 1d and Extended Data Fig. 3d). Additional monitoring effort has focused on fish, marine species and insects (Appendix 1, Supplementary Information; all supplementary materials are available at https://doi.org/10.5281/zenodo.8417583).

Turkey is by far the largest COST country by area, and with almost 784,600 km², it is 42% larger than the next largest country, France (excluding its overseas territories). With no documented PGD monitoring, Turkey is an outlier for its size and absence of GME and is an influential observation in statistical analysis. When Turkey is omitted, analysis of the other COST countries demonstrates that larger countries tend to have higher GME (Fig. 2a; negative binomial regression,

$P = 0.02$). In contrast, intermediate GDP is associated with greater GME (Fig. 2b; negative binomial regression, GDP quadratic term $P = 0.003$; Appendix 2, Supplementary Information). Substantial residual variation remains, with Austria, Finland, Norway and the United Kingdom among those countries having fewer projects than expected, and Belgium and Sweden more projects, in relation to both size and GDP (Fig. 2b). The negative quadratic relationship of GME with GDP remains statistically significant with the omission of data from any single outlier or extreme value (Switzerland, Ireland or Luxembourg; Appendix 2, Supplementary Information).

**Joint environmental niche marginality framework**

To integrate PGD monitoring into a framework for addressing climate change impacts, we evaluated the relationship between GME and expected climate change effects on species' trailing-edge climatic niche marginality. These areas correspond to the portion of the least suitable 25% of niche conditions that becomes less suitable with climate change (see the Methods for a full description). The four groups of study species consist of amphibians (44 Anura and 26 Caudata), large birds (16 species in the Accipitridae, Anatidae, Gallidae and Otididae), carnivorans (8 species) and forest trees (91 species), a total of 185 species. The species were chosen for their current or potential future conservation or management interest (Extended Data Table 1) and, except for carnivorans, generally reflect the trend towards greater species richness in southern Europe and the Mediterranean region[32] (Extended Data Fig. 5). We calculated the values of an index of climate

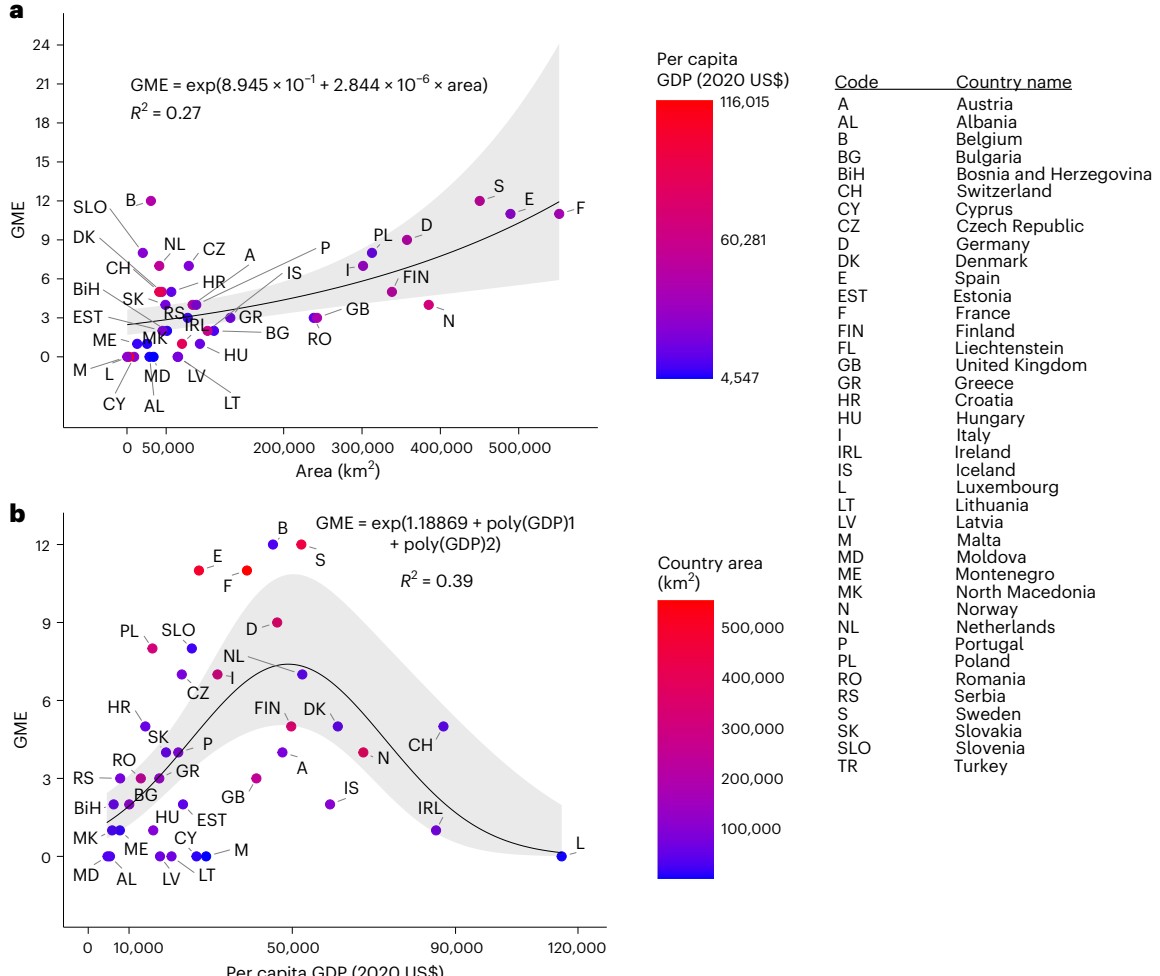

**Fig. 2 | GME of COST full-member countries as a function of area per capita GDP. a,b,** Generalized linear models for the GME of COST full-member countries, represented by international postal codes, as a function of area (**a**) and per capita GDP (**b**). The equations of the lines are shown, along with 95% confidence intervals in shading. The models were fit as negative binomial distributions with the log link function. Model fit is given as Veall-Zimmermann $R^2$. Turkey is of substantially greater geographic extent than the displayed countries, but it

has no documented GME and is omitted as an outlier and influential observation. Both the linear area term and the quadratic GDP term are significant in the multiple generalized model corrected for spatial autocorrelation (two-tailed tests; area: $z = 2.269$, $P < 0.0233$; GDP quadratic: $z = -2.969$, $P = 0.00299$; see the Methods for the details). A significant quadratic term remains upon the omission of any one of the three high-GDP countries.

niche marginality[33] separately for each species and for each pixel within its global range, on the basis of range-wide climate. Pixels with the highest 25% of index values in the species' range globally indicate the geographic distribution of climatic niche marginality for that species. Current and future distributions of niche marginality for the species in the four study groups are diverse and complex (Appendices 3–6, Supplementary Information). For example, changing spatial patterns of niche marginality and core niche conditions of the Swiss stone pine (*Pinus cembra* L.) produce a geographic mosaic of changing environmental suitability (Fig. 3). The degree of change of niche marginality and the loss of suitable climatic conditions within the species' current range depend on the severity of predicted climate change, a pattern seen in many other species (Fig. 3b versus Fig. 3c and Fig. 3d versus Fig. 3e; see also Appendices 3–6, Supplementary Information).

We superimposed the areas of species' niche marginality at the trailing edge to identify areas where climate change will negatively impact many species. Increases and decreases in the total number of study species with populations at niche margins vary broadly across COST countries but are similar between climate change scenarios (compare Fig. 4a–d). Assuming that species' climatic niches remain stable over time, we predict increases in the number of species with

marginal habitat in Austria, Bulgaria, Germany, Hungary, Poland and Romania. Countries in central and eastern Europe also hold relatively many species that newly experience marginal niche conditions in the future period (Fig. 4e,f). More species lose areas of suitable climatic conditions in southern European countries than elsewhere in Europe (Fig. 4g,h). These trends are similar in other combinations of global circulation model (GCM) and Shared Socioeconomic Pathway (SSP) (Appendix 7, Supplementary Information).

Spatiotemporal trends in niche marginality in the four groups of species also differ substantially among COST countries (Extended Data Fig. 6 and Appendix 7, Supplementary Information). Across the four groups, the number of species with habitat at niche margins in each country is similar between current and future periods (Extended Data Fig. 6, left column versus middle column). Nonetheless, the number of species of amphibians with climate conditions at niche margins increases in central and eastern Europe (Extended Data Fig. 6a–c), as does the number of large bird species with niche margin conditions, especially in France, Spain and Italy (Extended Data Fig. 6d–f). We predict that the number of carnivorans that experience climates at trailing niche margins will increase in some Nordic countries and in central Europe (Extended Data Fig. 6g–i), providing no evidence of

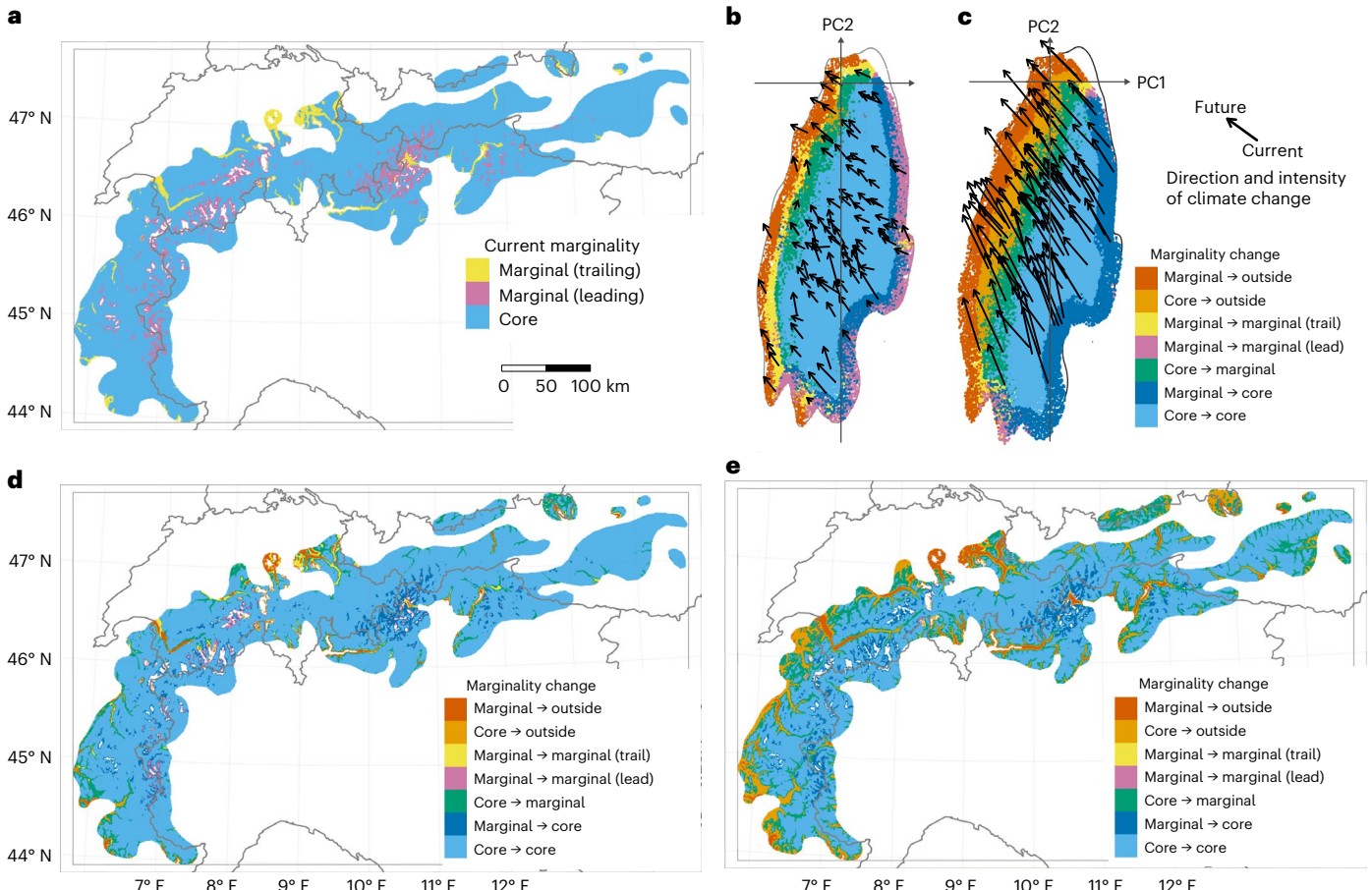

**Fig. 3 | Current and future climatic niche marginality for the Swiss stone pine (*Pinus cembra*) by 2041–2070. a**, Current marginal and core areas. Marginal areas are split into trailing and leading edges on the basis of differences between current and future Niche Margin Index (NMI) values: NMI values increase at the leading edge; NMI values decrease at the trailing edge. **b**, Predicted changes in climatic marginality according to the mildest climate change scenario (MPI-ESM1-2-HR, SSP 3-7.0). All pixels of the range of the species are represented in the principal component (PC) analysis space with coloured points corresponding to categories of possible changes in climatic marginality. The direction and intensity of change in climatic conditions for a random subset of 100 pixels are indicated with arrows. **c**, Predicted changes in climatic marginality according to the harshest climate change scenario (UKESM1-0-LL, SSP 5-8.5). **d**, Changes in climatic marginality categories across the range of the species for climate change scenario MPI-ESM1-2-HR, SSP 3-7.0 (geographic representation of **b**). **e**, Changes in climatic marginality categories across the range of the species for climate change scenario UKESM1-0-LL, SSP 5-8.5 (geographic representation of **c**).

a north–south trend in changing niche marginality in Europe for this taxon. The data suggest that the number of tree species experiencing niche margin conditions will increase broadly across central and northern European countries (Extended Data Fig. 6j–l).

Regional differences in niche marginality are also visible at the pixel level (Fig. 5 and Extended Data Fig. 7), at which national trends are less apparent. Patterns of current joint niche marginality vary among the four study groups, with foci of joint niche marginality in the Iberian Peninsula and the eastern Adriatic coastline (amphibians and forest trees; Fig. 5a,b,g,h); in the Iberian Peninsula, the Alps and central Turkey (large birds; Fig. 5c,d); and in several restricted areas broadly across Europe (carnivorans; Fig. 5e,f and Appendix 7, Supplementary Information). Current joint niche marginality estimates for different climate scenarios are largely similar (Fig. 5 and Appendix 7, Supplementary Information). The loss of suitable climatic conditions under relatively severe climate change (for example, GCM UKISM1-0-LL/SSP 5-8.5, files with 'loss' in the name, Appendix 8, Supplementary Information) leads in the future to areas in which joint niche marginality is reduced (for example, amphibians, Extended Data Fig. 7a–c; trees, Extended Data Fig. 7j–l). Comparison of current and future distributions of niche marginality in individual species often indicates the conversion of core conditions to marginal ones, but not always in the southern portion

of species ranges (Appendices 3, 4 and 6, Supplementary Information). Many amphibian and tree species are endemic to Europe, have restricted ranges and lose current areas of suitable conditions with changing climate, including the loss of both niche marginal and core areas. This is analogous to the areas coloured red and orange in Fig. 3 (Extended Data Fig. 7a–c,j–l and Appendices 3, 6 and 8, Supplementary Information). Such losses also occur for some large European birds, especially *Aquila adalberti* (Appendix 4, Supplementary Information). Additional species losing substantial portions of both current trailing-edge marginal and current core niche conditions include *Abies pinsapo*, *Alnus cordata, Alytes cisternasii* and *Lynx pardinus* (Appendices 5 and 6, Supplementary Information). In contrast, species with only a small portion of their range in COST countries, such as wolverine (*Gulo gulo*) and brown bear (*Ursus arctos*), show little change in the distribution of habitat areas having marginal climatic niche conditions in COST countries (Appendix 5, Supplementary Information).

The relationship between future joint niche marginality in each country and country GME varies greatly but shows no linear relationship (Fig. 6a). The results are similar when monitoring effort is restricted to terrestrial species only (Fig. 6b), and the order of country values is little affected by choice of GCM and SSP (Appendix 9, Supplementary Information). Countries exhibiting both few study species

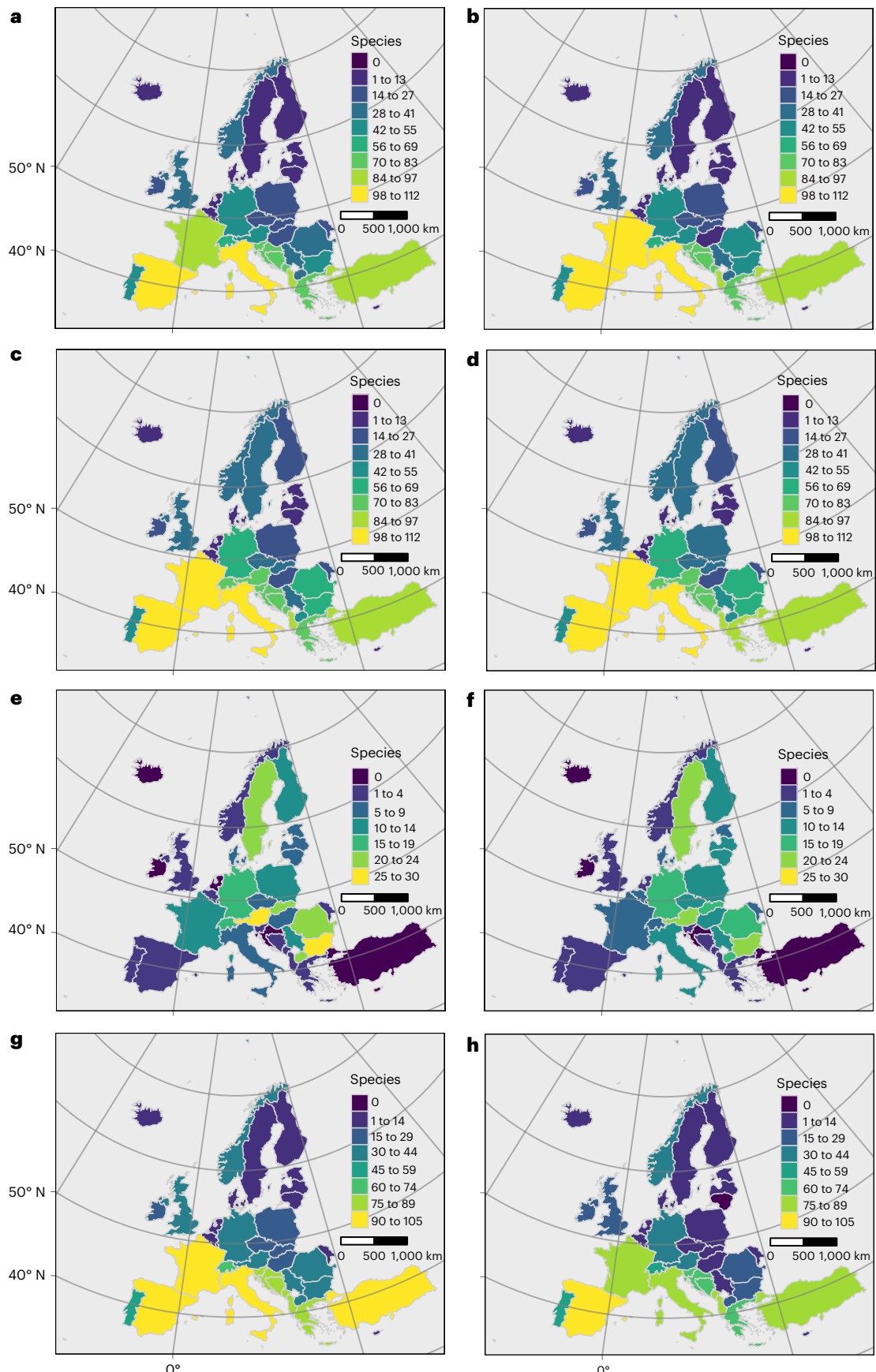

**Fig. 4 | Study species with trailing-edge marginal niche conditions, by country. a,b**, Current trailing-edge marginality. **c,d**, Future trailing-edge marginality. **e,f**, Species newly experiencing marginal habitat. **g,h**, Species experiencing habitat loss. The left panels show the results under harsh climate change (GCM UKESM1-0-LL, SSP 5-8.5); the right panels show the results under mild climate change (GCM MPI-ESM1-2-HR, SSP 3-7.0).

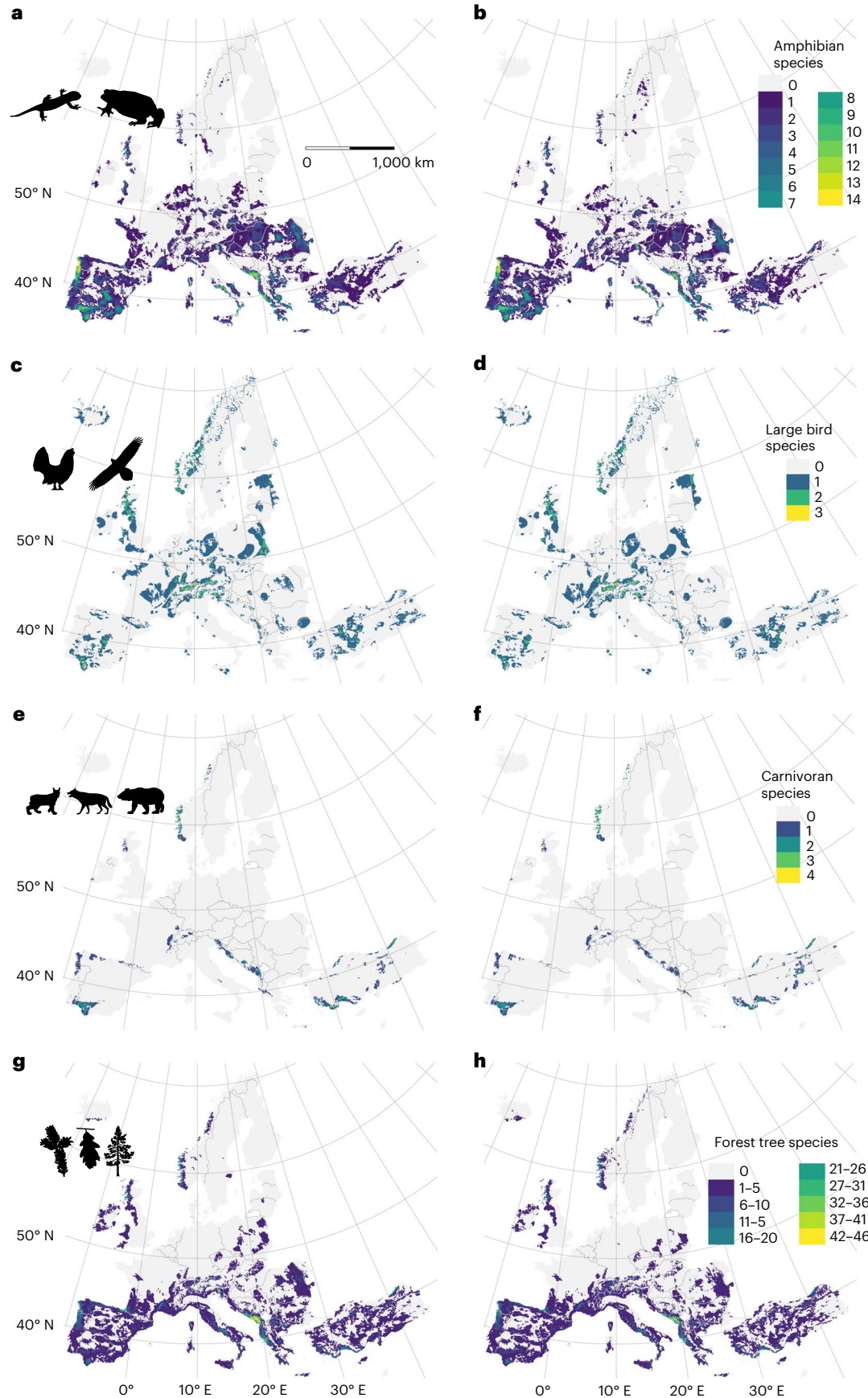

**Fig. 5 | Current species' joint niche marginality. a–h,** The colours represent the pixel tally of species with trailing-edge niche conditions. The left panels show the results under severe climate change (SSP/GCM: 5-8.5/UKISM1-0-LL). The right panels show the results under milder climate change (3-7.0/MPI-ESM1-2_HR). The data represent 70 amphibian species (**a,b**), 16 birds (**c,d**), 8 European carnivorans (**e,f**) and 91 tree species (**g,h**).

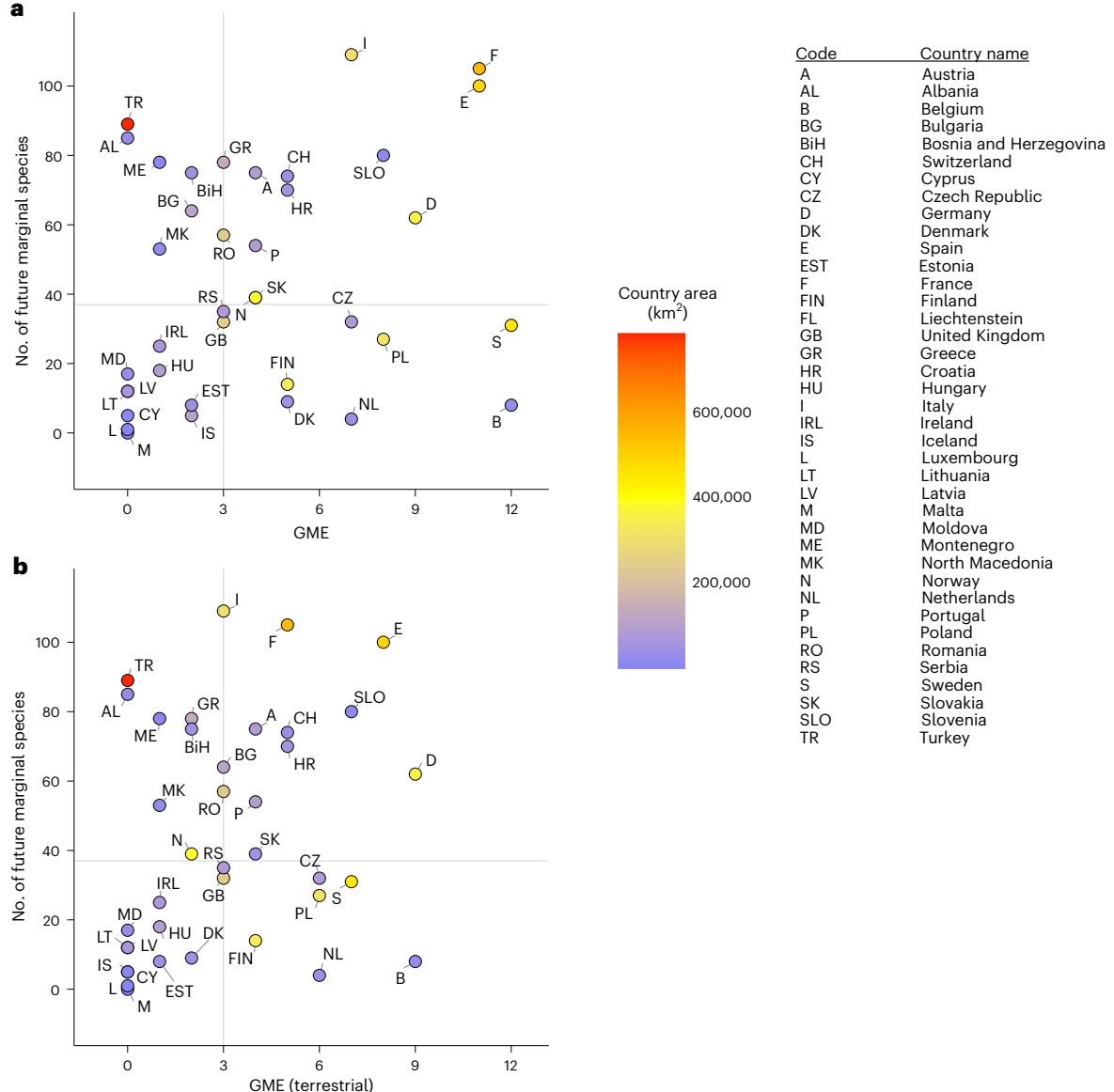

**Fig. 6 | The relationship between GME and the number of species with marginal climatic niche conditions as of the years 2041–2070. a**, All Category II monitoring as an indicator of effort at the national level. **b**, Programmes to monitor genetic diversity in selected amphibian, avian, carnivoran and plant species only. Countries are indicated by postal codes. Marginal species include all species chosen for the calculation of marginality, including non-troglobite amphibians, a collection of large birds, selected large carnivorans and a set of forest trees. No general linear trends exist, although there is substantial variation both in numbers of species in marginal niche situations and in GME of countries.

with trailing-edge niche conditions and relatively little monitoring effort (lower left quadrant in Fig. 6) are of relatively small geographic area, although many smaller countries are not in this quadrant. The results for species' current marginality versus monitoring effort are similar to these, as are the results for change in the number of marginal species between the two periods versus effort (Appendix 9, Supplementary Information). Variation among countries in the degree of change in the number of species with trailing-edge marginal conditions between the current and future periods is broadly distributed across Europe (Extended Data Fig. 8). The results from other GCM–SSP combinations are similar (Appendix 7, Supplementary Information).

## Discussion

Contrary to our expectations, the areal extent of countries does not generally account for variation in GME. Only by excluding Turkey as an outlier did we observe a positive relationship between country area and GME. Turkey produces population genetic research but is not a member of the European Union (EU). The reporting requirements of the EU Habitats and Birds Directives may successfully promote the use of Category II genetic monitoring. In contrast, and in line with our expectations, countries with relatively low per capita GDP generally have lower GME. However, it appears that countries with intermediate GDP have on average the highest GME. Countries with high GDP are in many cases relatively small (Fig. 2), and many factors conceivably influence the establishment of monitoring programmes, regardless of country size or per capita GDP (such as the number of wild species of traditional or cultural importance, or species richness). Extensive exploration of country characteristics that influence the establishment of programmes for monitoring PGD is beyond the scope of this paper but could be explored in future research, perhaps using data from country reports to the CBD on progress in the implementation of the Kunming-Montreal Global Biodiversity Framework[1,4].

The monitoring programmes we report here generally focus on detecting changes in population diversity of neutral nuclear marker

loci and of mitochondrial DNA (haplotypes). These loci are probably not directly involved in adaptation to climate. The studies minimally report allelic or haplotype diversity, and none are specifically designed to detect genetic responses to climate change or deteriorating environments per se. Genetic characteristics of populations at environmental niche margins could make these populations critical resources for managing the impacts of climate change, such as through translocation programmes[36,37] (but see ref. [38]). However, monitoring neutral genetic markers and indicators of effective population size alone is unlikely to provide representative data on the ability of populations to adapt to changing environments (for example, caused by ongoing climate change), because of weak correlations between population genetic marker loci and specific genetic variants affecting functional traits that confer adaptation to the environment[39–41]. Furthermore, the ability of monitoring studies to characterize adaptive potential via measures of genome-wide diversity and/or niche marginality is an ongoing area of research[42–44]. Nonetheless, GME and genetic monitoring using marker loci are suggestive of the future capacity of countries to conduct monitoring of genetic diversity related to predicted or observed climate change responses of species. Countries with large GME should be relatively well prepared to evaluate climate impacts on genetic diversity because they have the relevant infrastructure (that is, genetic laboratories) and experience.

Efforts to increase capacity for genetic monitoring could emphasize southeastern COST countries, where the number of species in areas at climatic niche margins is currently relatively high and expected to remain so in the future (Figs. [4] and [5] and Extended Data Fig. 6). Here, GME for terrestrial species is sparse (Extended Data Fig. 2). These conditions suggest that certain countries (upper left in Fig. [6]) present relatively high opportunity/need for climate-guided genetic monitoring and relatively low GME historically. Some commonalities notwithstanding, the areas where species will probably experience environmental deterioration differ depending on the taxonomic group under consideration (Extended Data Fig. 6c,f,i,l). Baseline genetic assessments are needed in some geographic areas, such as the Iberian Peninsula, Italy and France for amphibians and southeastern Europe for forest trees (Extended Data Fig. 6 and Appendices 3 and 6, Supplementary Information), where multiple species will experience environmental deterioration due to rapidly changing climate, as shown by patterns of joint niche marginality (Extended Data Fig. 7c,l). Our results, here based on a joint niche marginality approach, indicate that for various groups of species, the Iberian Peninsula, the eastern Adriatic coast, central Turkey and the Carpathian Mountains can serve as foci for international, cooperative monitoring programmes that anticipate the effects of climate change by establishing genetic baselines that include populations in these areas.

To address the importance of environmental gradients to the conservation of genetic diversity, we distinguish here between populations that are geographically peripheral with regard to a range centroid and populations that are environmentally marginal, occurring towards the limit of their realized environmental niche. Relative geographic position can present little relationship to variation at functional loci, while relative environmental niche marginality of populations can predict variation at these loci and demographic events in populations near niche limits[41,45]. The establishment, adaptation and persistence of populations at environmental niche margins may depend on baseline genetic diversity, the steepness of environmental gradients, the rates of gene flow from non-marginal populations and stochastic processes[20,46]. Monitoring projects should estimate changes in genome-wide diversity when sufficient material and financial resources are available[42] and should span environmental gradients to include populations from both core and marginal niche situations. Such studies will help elucidate how genetic diversity and adaptive potential vary across species ranges and respond to climate change, something that is not possible without a temporal component in the sampling and analysis[47,48].

The distributions of some species may be more limited by anthropogenic factors than by climate, such as for some large carnivores. Species' climatic tolerances may therefore not be well estimated by our methods. For example, the Iberian lynx (*Lynx pardinus*) may lose less area of suitable climatic conditions than estimated here (Appendix 5, Supplementary Information). Examination of these patterns is left for future studies that take focal-species approaches. Furthermore, range expansion with climate change will result in the influx of species into areas with newly suitable climate on leading range edges[16]. In addition, taxonomic revisions (for example, splitting) can change the niche breadth of the revised taxa, with resulting changes in the geographic distribution of areas of niche marginality. Follow-up studies can refine predictions for climatic conditions and niche marginality in the context of taxonomic revisions and specific goals for genetic monitoring and population management. Category II monitoring programmes that span climatic gradients occupied by focal species should be established in additional countries not involved in the COST programme, wherever climate analysis suggests increasing niche marginality of populations of conservation interest, or wherever a risk of genetic erosion is suspected.

Detecting any loss of genetic diversity in niche margin populations should be a priority, and if detected, it should probably trigger a management response. To inform management in this way, monitoring projects need to span entire environmental gradients as occupied by species, in order to sample relevant genetic variation in niche marginal populations. Genetic samples from such prospective monitoring designs will be well suited for evaluating PGD and the adaptive capacity of populations, and for designing appropriate management strategies[49]. Our results may guide future EU investment in genetic monitoring programmes and in conservation genetics/genomics research projects. Positive developments in the support of PGD monitoring that have arisen from the 15th Conference of the Parties to the CBD can be leveraged by adopting language in the EU's Birds and Habitats Directives to support genetic monitoring. Similar actions should be taken by governments outside of Europe. Future projects may productively focus networking and training efforts more strongly in certain regions, such as the Balkan countries and Turkey.

## Methods

We compared data on GME and climatic niche marginality to address whether historical effort and experience in PGD monitoring at a national scale correspond to the anticipated impacts of climate change on environmental suitability for ensembles of wild species. We call this approach a 'joint niche marginality' framework to express how areas of marginal conditions within the niches of multiple species coincide geographically, and we used it to propose taxonomic and geographic foci for future programmes of genetic monitoring. To address our four guiding questions, we report results from a comprehensive survey of the scientific literature, as represented in the Web of Science Core Collection of journals, with use of a simple, inclusive search string of relevant terms. We also collected references and documentation of unpublished monitoring programmes by using professional networks to comprehensively access the grey literature, including governmental and non-governmental reports and web pages in national languages. We focused our analysis exclusively on monitoring programmes that report repeated measures of PGD indicators that were developed with molecular genetic or genomic tools (Category II programmes[34]), and we excluded genetic assessments, which lack temporal replication, from consideration. We compiled and summarized these data by country to address the geographic and taxonomic distribution of monitoring projects as an indicator of GME. We then assembled groups of species of current or potential conservation interest on the basis of taxonomic and functional characteristics and predicted changes in their environmental niche marginality within their current range by using the range-wide occurrence of species, range polygons, and digital land

cover and climate layers, the latter of which express current climate and projected changes[33].

### Distribution of GME in Europe

**The grey literature.** Beginning in October 2019, we began to solicit the submission of published and unpublished (grey literature) materials documenting genetic monitoring programmes, projects and activities (hereafter 'projects'). We used social media and e-mail to contact the extended network of relationships centred on participants in the COST Action 'Genomic Biodiversity Knowledge for Resilient Ecosystems' (G-BiKE, https://www.cost.eu/actions/CA18134/), a Europe-wide effort to improve and promote the use of genetic and genomic methods for supporting the delivery of ecosystem services. We directly contacted colleagues, government officials and non-governmental agency representatives in their home countries to identify and solicit information on past and ongoing projects. Submission of the requested information (Table 1) was open to this broad community of scientists, policymakers and stakeholders and was structured by variables describing each project, organized in an online spreadsheet (Appendix 11, Supplementary Information). We laboured to follow leads and make direct contacts to obtain internal documents and unreleased private reports. We collected all available documentation in the form of web documents and their URLs, white papers, internal and released reports, and published papers that were associated with and substantiated each submitted project. Solicitation and submission of information continued until 31 December 2021. We focused our data collection efforts exclusively on COST full-member countries (hereafter, COST countries) except for those entering COST after the end of data collection: Ukraine, Georgia (31 March 2022) and Armenia (10 November 2022). Submitted projects that did not sample populations in at least one COST full-member country were excluded from subsequent data aggregation and analyses.

We developed standardized criteria for judging the validity of projects to monitor PGD by following a published definition of genetic monitoring[34] and by defining a decision tree (Extended Data Fig. 9). Each submitted project was assigned using computer-generated pseudo-random numbers to 2 of 14 evaluators, who sought additional information in national languages as needed through web search and personal inquiries. Pairs of evaluators examined projects independently from one another. When the evaluators disagreed on project validity, they attempted to reach consensus. Persistent disagreements were mediated by two co-authors (P.B.P. and M.B.). Written documentation, broadly defined, was required for a positive decision on project validity, thereby excluding projects reported only by personal communication or e-mail or lacking documentation (Extended Data Fig. 9). Valid monitoring projects included those that acquired and analysed genotype data from the same populations or identical locations, at two or more time points at least one year or one generation apart, whichever was longer. Additionally, candidate projects needed to explicitly declare the goal of informing management and/or conservation policy and activities (Extended Data Fig. 9). These criteria excluded studies lacking temporal replication, studies on pathogens and disease vectors, and studies focused on questions clearly restricted to the field of population biology and without explicit conservation motivation.

A second round of evaluation classified valid monitoring projects into two groups. We distinguished between Category I projects, which collected genotype or haplotype data for species and individual identification, and Category II projects, which reported at least one index of PGD, such as number of alleles, observed or expected heterozygosity, etc.[34]. The use of genetic data from archived samples or collections to establish an initial temporal reference for focal populations was acceptable, as long as the populations were strictly identical. Certain problems were presented by projects that evaluated changes in genetic diversity in reintroduced populations and those receiving introduced individuals to support levels of PGD (that is, genetic support or assisted gene flow)[36]. For the validity of these studies

as Category II monitoring, a baseline sample was needed from the population of individuals initially chosen for reintroduction, or repeat temporal samples from the focal, reintroduced or supported population itself. We excluded projects comparing genetic diversity in contemporary samples to that from the original or putative source populations when these were sampled only after (re)introductions, due to the potential for sampling bias. As in the initial evaluation of validity, both evaluators needed to express a consensus concerning the type of monitoring (Category I or II) that was conducted.

**The scientific literature.** We also conducted a separate survey of the peer-reviewed scientific literature to identify projects monitoring genetic diversity. On 1 December 2021, one co-author (P.B.P.) conducted a search of all Web of Science collections with the search string "Topic: 'genetic population diversity monitoring' NOT 'cell' NOT 'virus' NOT 'medical'". The citations were then filtered to come only from the following journal categories: Agriculture, Agronomy, Dairy Animal Science, Biodiversity Conservation, Marine Freshwater Biology, Ecology, Entomology, Environmental Sciences, Evolutionary Biology, Fisheries, Forestry, Genetics and Heredity, Horticulture, Multidisciplinary, Multidisciplinary Sciences, Ornithology, Plant Sciences and Zoology. Other strategies, such as additionally restricting the search to COST countries, resulted in the omission of studies that qualified as Category II monitoring in Europe. One co-author (P.B.P.) scored all collected citations for being conducted in COST countries and for being either Category I or II monitoring. Each of these candidate studies was re-examined independently by one of four additional co-authors (D.R., E.B., A.K. and F.E.Z.) to evaluate the initial assessment and to identify redundancy within the original list of validated projects. Confirmed, non-redundant cases were then added to the list of monitoring projects. Ad hoc repetition of the Web of Science search to identify additional studies published in late 2021 and efforts to obtain documentation of specific unpublished projects, produced before the end of 2021, continued during the first four months of 2022.

We focused on Category II monitoring studies because of their relevance to mandates to conserve genetic diversity, and we carefully tallied these studies by country and by taxonomic and additional groupings (Appendix 11, Supplementary Information). We considered submitted projects that monitored particular single species in a country as distinct projects when different populations were studied by different research groups, institutes or organizations. We also considered projects conducted by a single research group but having more than one focal species as distinct. Projects addressing different focal populations of a single species, analysed as exclusive, distinct sets of populations by a single research group, were also counted as distinct projects. Nonetheless, publications that presented analyses of repeated samples from a single set of populations, and were extensions of original studies and used the original published data in establishing temporal trajectories of genetic diversity, were not counted as separate projects regardless of author identity. Analyses of samples by contract laboratories, in a separate country from that of the study population(s), research group or monitoring organization, did not qualify the project to count towards the tally of projects for that separate country, unless of course at least one sampled population came from that country. In multi-country projects generally, samples for genetic analysis needed to be physically collected within a country for a project to count towards the tally of projects in that country. This meant that potentially a project was assigned (tallied) only to a subset of participating countries, those that were the sources of genetic samples. Projects reporting a temporal trajectory of genetic diversity in captive or domestic populations needed to employ genetic analysis of repeated samples and not rely exclusively on estimates of genetic diversity or change thereof that were obtained from pedigree analysis of breeding records. Because some projects sampled populations in more than one country, we defined the GME of a country as the tally of Category II monitoring projects

obtaining genetic data from within the country. We determined the geographic distribution of GME for focal taxonomic and functional species groups by mapping GME for each group in each COST country and examining the frequency distribution of GME among countries. We focus our analyses exclusively on Category II monitoring studies and will address Category I studies in a future publication.

## Climate niche marginality in Europe

**Focal species.** We defined four divergent groups of species for the examination of current and future geographical patterns of climatic conditions. Our objective was to construct groups with membership that exceeded the scope of current GME and that, because of taxon identity or life history traits, either are currently of conservation interest or could conceivably become of interest as climate change proceeds. Thus, while many of the species may be on national Red Lists in European countries, this was not a requirement for inclusion. We also did not attempt to comprehensively include species of conservation interest. We explicitly disregarded membership on Red Lists and EU Directives as criteria because the varying completeness, taxonomic resolution and criteria for species' inclusion of national Red Lists across Europe made it impossible to implement a single standard. Furthermore, not all COST countries are members of the EU and subject to the Directives. We developed lists of focal taxa to include (1) most native European Amphibia (44 Anura and 26 Caudata), because of their recognized sensitivity to climate change (we excluded cave-dwelling amphibians because of their limited exposure to terrestrial climate); (2) 16 species of large birds, representing the Accipitridae, Anatidae, Gallidae and Otididae, because size is related to extinction probability in birds globally[50]; (3) a set of 8 relatively large carnivorans because of their general economic, ecological and cultural importance; and (4) a set of 91 species of forest trees (64 Magnoliopsida and 27 Pinopsida), because of the general economic and cultural importance of trees (Extended Data Table 1). We focused on these groups because the range maps for the species are probably reliable, and the occurrence data are probably well reported. Global range maps for each focal species were retrieved as polygons from the data portal of the International Union for the Conservation of Nature (IUCN)[51], and species occurrence data were retrieved from the Global Biodiversity Information Facility[52-55]. We then defined species' global distributions as the pixels occupied by the species according to the IUCN range maps. We refined species' distributions within range polygons by filtering out pixels corresponding to CORINE Land Cover 2018 habitat classes[56] that were not intersected at least once by occurrences of the species in question. This removed urban areas and other habitat/land-cover types for which we found no evidence of occupation in the occurrence data.

**Marginality calculations.** We used the worldwide 19 bioclimatic variables from the Chelsa database of global climate values for the period 1981–2010 at 30 arcsec resolution (http://chelsa-climate.org[57]) to calibrate principal component scores. We defined a working environmental space consisting of the first two principal component axes. This space summarized the main climatic gradients present on Earth (75.7% of the variation explained). We rasterized the IUCN species range maps at 30 arcsec resolution, extracted bioclimatic values for every occupied pixel (after filtering with CORINE 2018) and projected these values to the global climate space to generate species scores[58]. Using these scores, we delineated the niche margins of each species by kernel density estimation (that is, the 0.99 quantile)[33,58]. These margins described the boundaries of the climatic conditions currently occupied by the species globally. Finally, we calculated the Niche Margin Index (NMI), a standardized metric of climate marginality, for each pixel of each species distribution, on the basis of the multivariate distance to the niche margins and using the approach of Broennimann et al.[33]. The NMI metric for each species varies from 0 to 1, with values of 0 indicating that the climatic conditions in the pixel are at the niche margin

and values of 1 indicating conditions at the niche centre. To provide synthetic niche marginality maps for each species, we considered that pixels with the 25% most marginal conditions for a species (NMI < 0.25), determined globally, constituted climatically marginal areas for the species, while the rest of the pixels within the species' niche constituted the core conditions. In this way, species' niche marginality scores translated directly from a multivariate space to a geographic distribution within the current range of the species (for example, Fig. 3). Notably, niche marginal situations can occur in both geographically central and peripheral areas within the species range.

To map the future distribution of marginality of species' climate niches, we updated the climatic values of pixels corresponding to the species distributions within the study area using two SSP scenarios, the relatively moderate SSP 3-7.0 (regional rivalry) and the worst-case SSP 5-8.5 (fossil fuel development). We chose these scenarios because growth in carbon emissions currently is not showing evidence of moderation[59]. We chose three GCMs, IPSL-CM6A-LR, UKISM1-0-LL and MPI-ESM1-2_HR, to incorporate substantial variability among GCM models in the analysis. This provided six combinations of SSP scenarios and GCMs. We then obtained the simulated climate data for a baseline period, 1981–2010, and for a 30-year future period, 2041–2070, from the Chelsa database v.2.1 (ref. 60). We recalculated the NMI metric for each species in each pixel to produce maps of species' future niche marginality and the transition of areas between core and marginal niche categories for each SSP–GCM combination. We identified leading and trailing niche marginal areas by examining how NMI values changed over the time interval: NMI increases for leading edge pixels (conditions move closer to the niche centroid), while trailing edge pixels exhibit decreasing NMI values (conditions move further from the niche centroid). The number and distribution of trailing-edge pixels and changes in NMI values can vary among the SSP–GCM combinations. Mapping the geographic distribution of future niche marginality in this way assumes that the climate niches of species fulfil the assumptions of niche stability[61,62] and niche filling[63]. Multispecies marginality maps were produced for each species group by stacking the species' marginality maps and calculating maps of the number of species in marginal niche conditions for each pixel, at present and in the future. We compared maps of current and future niche marginality to identify pixels in which we estimated that populations of species will shift into climatically marginal niche conditions in the future, and other changes determined by changing NMI values.

To facilitate the comparison of GME to the predicted effects of climate change on species' niche situations at the country level, we converted species maps of niche marginality to country tallies of species with trailing marginal niche conditions and tallied change over time. For each COST country, we obtained a shapefile of country boundaries at 10 m resolution from the Natural Earth website[64]. We excluded overseas territories and regions of European countries—that is, islands and areas outside of a rectangular bounding box defined by 25° W, 57° E, 29.1° N and 73° N. This excluded, for example, the Canary Islands (Spain), Svalbard (Norway) and French Guiana (France). We used the R package tmap[65] to map the number of PGD monitoring projects in each country, the number of marginal species in each focal taxonomic group currently and in the future, the predicted number of species that newly experience niche margin conditions within a country and the count of species that lose suitable niche conditions. We plotted counts of species, using current and future joint niche marginality, and their change between periods against country tallies of PGD monitoring programmes. These plots allow visualization of the relationship between national effort for PGD monitoring and geographic foci of future climatic niche margin conditions for multiple species.

## Statistical analyses

We compared GME among countries by modelling the number of Category II monitoring projects as a function of two national indicators.

We used country area as an example indicator of the physical aspects of countries, and we estimated the land area of COST countries in continental Europe, in the Mediterranean and Baltic islands, and in Asia using the R package sf[66]. While many more physical aspects could be explored, a comprehensive study of these aspects of COST countries is beyond the scope of the paper. We also chose per capita GDP as an example indicator of economic activity and available resources, one that is available for all COST countries. Data on GDP in 2020 US dollars were obtained from an authoritative online source[67], the most recent year for which data from all COST countries were available. The relationship of monitoring effort with many other social and economic indicators could be explored, but we leave this as well for future analyses. On the basis of inspection of scatter plots, country area entered the models as a first-order effect, while GDP entered as a second-order orthogonal polynomial.

We used a generalized linear model (GLM) framework to analyse country counts of PGD monitoring projects. Models were fitted with functions from the R packages stats, MASS and hermite[68,69]. Outlier and influential data points were identified with leverage statistics and by inspection. We quantified model explanatory capacity with the Veall–Zimmermann[70] pseudo-$R^2$ calculated on deviance residuals, and we used model likelihoods and $\chi^2$ statistics to compare models during model development. We modelled the data with Poisson, negative binomial and Hermite regressions and based statistical decisions on negative binomial models because of a significant reduction in over-dispersion of residuals in comparison with the Poisson model, and no additional improvement provided by the Hermite model (Appendix 2, Supplementary Information). We examined negative-binomial GLM residuals for small-scale spatial autocorrelation (SAC) using a randomization test of the significance of Moran's $I$ ($H_0$: $I = 0$), at successive intervals of 300 km between country centroids, using the correlog function in the R package ncf[71]. We did not address large-scale spatial structure (>1,500 km). Because SAC can bias tests of significance of model effects when analysing spatial data, we removed SAC from GLM residuals by first constructing spatial eigenvectors (Moran's Eigenvector Maps) with the function mem from the R package adespatial[72] and a regional distance network among country centroids constructed with the functions dnearneigh and nb2listw in the R package spdep[73] (Appendix 2, Supplementary Information). Eigenvectors with positive eigenvalues were included as additional linear terms (regardless of statistical significance) in the GLMs. We added eigenvectors until the $P$ values of the randomization test of Moran's $I$, calculated on model residuals at intervals to 1,500 km, equalled or exceeded 0.05 after rounding. Although significance levels were reduced by the addition of spatial eigenvectors, decisions concerning the statistical significance of model terms were not affected. All tests of significance were two-tailed. Mapping and statistics were conducted in R version 4.1.2 (ref. 74).

### Reporting summary

Further information on research design is available in the Nature Portfolio Reporting Summary linked to this article.

## Data availability

The raw data on submitted candidate monitoring projects and a variable indicating their validity as Category II monitoring are available as Appendix S11 and for download at https://figshare.com/s/296e3bf1db7b84ec71bd. All data used in the study are available in compressed archives in a Zenodo repository[75] at https://doi.org/10.5281/zenodo.8417583. Source data are provided with this paper.

## Code availability

The code to produce all graphics and statistical analyses, along with the necessary raw and intermediate data, can be found in compressed archives in a Zenodo repository[75] at https://doi.org/10.5281/zenodo.8417583.

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

## Acknowledgements
This publication is based on work from COST Action G-BiKE, CA 18134, supported by COST, www.cost.eu. I.P.-V. was supported by the US Geological Survey Powell Center for Synthesis and Analysis. B. Rolečková was supported by INTER-EXCELLENCE—INTER-COST (LTC20021). D.P. and L.L. were supported by grants to L.L. from the Swedish Research Council Formas (grant no. 2020-01290) and the Swedish Research Council (grant no. 2019-05503). T. Aavik was supported by Estonian Research Council grant PRG1751. A.B. was supported by the Slovenian Research and Innovation Agency (program group P1-0386). A.F. acknowledges support from the Romanian Ministry of Research, Innovation and Digitalization funds PN23090304 (12N/01.01.2023) and CresPerfInst (34PFE/30.12.2021). A.K. was supported by Norges forskningsråd (the Research Council of Norway), grant no. 160022/F40 NINA. K.K.S. and M.W. acknowledge support from the Slovenian Research Agency (research core funding no. P4-0107). P.B.P. acknowledges support from grant no. PID2020-118028GB-I00 funded by MCIN/AEI 10.13039/501100011033. All silhouettes in the figures are from PhyloPic. This paper is dedicated in memory of our colleague and friend Michael Bruford (1963–2023).

## Author contributions
M.B. and P.B.P. conceived the study. P.B.P. conducted a search of the published literature, organized and mapped the monitoring project data, and conducted the statistical analyses. O.B. conducted the niche marginality analysis and mapping with support from A.G. P.B.P., P.C.A., L.D.B., A.B., E.B., V.C.-C., J.A.G., C.H., P.K., M.K.K., A.K., C.N., D.P., B. Rolečková, D.R., K.K.S., S.T., C. Vilà and M.B. evaluated the submitted monitoring projects. T. Aavik, P.C.A., F.A.A., A.B., M.B., E.B., V.C.-C., M.D., A.F., A.P.F.-P., B.F., F.G., R.H., C.H., L.I., B.K.S., P.K., M.K.K., A.K., L.L., J.M., S.P., P.B.P., D.P., C.R.P., J.A.M.R., B. Rolečková, K.K.S., I.N.T., N.V., P.V., M.W. and F.E.Z. contributed data on validated Category II population diversity monitoring. P.B.P. wrote the first draft. T. Albayrak, S.H., M.L.-F., B.J.M., I.P.-V., L.S., G.S., M.S., B. Rinkevich, H.T. and C. Vernesi participated in discussions and contributed to the writing, as did all other co-authors. P.B.P and O.B. are co-first authors. A.G. and M.B. are co-senior authors.

## Competing interests
The authors declare no competing interests.

## Additional information
**Extended data** is available for this paper at https://doi.org/10.1038/s41559-023-02260-0.

**Correspondence and requests for materials** should be addressed to Peter B. Pearman.

Peter B. Pearman [1,2,3,61] ✉, Olivier Broennimann [4,5,61], Tsipe Aavik [6], Tamer Albayrak[7], Paulo C. Alves [8,9,10], F. A. Aravanopoulos [11], Laura D. Bertola [12], Aleksandra Biedrzycka[13], Elena Buzan [14,15], Vlatka Cubric-Curik [16], Mihajla Djan [17], Ancuta Fedorca[18,19], Angela P. Fuentes-Pardo [20], Barbara Fussi[21], José A. Godoy [22], Felix Gugerli [23], Sean Hoban [24], Rolf Holderegger [23,25], Christina Hvilsom[26], Laura Iacolina [27,28], Belma Kalamujic Stroil[29], Peter Klinga [30,31], Maciej K. Konopiński[13], Alexander Kopatz [32], Linda Laikre [33], Margarida Lopes-Fernandes[34,35], Barry John McMahon[36], Joachim Mergeay [37,38], Charalambos Neophytou [39,40], Snæbjörn Pálsson [41], Ivan Paz-Vinas [42], Diana Posledovich [33], Craig R. Primmer[43], Joost A. M. Raeymaekers [44], Baruch Rinkevich [45], Barbora Rolečková [46], Dainis Ruņģis [47], Laura Schuerz [23], Gernot Segelbacher [48], Katja Kavčič Sonnenschein [49], Milomir Stefanovic [17], Henrik Thurfjell [50], Sabrina Träger [51,52], Ivaylo N. Tsvetkov[53], Nevena Velickovic[17], Philippine Vergeer [54], Cristiano Vernesi [55], Carles Vilà [22], Marjana Westergren [49], Frank E. Zachos[56,57,58], Antoine Guisan [4,5,62] & Michael Bruford[59,60,62,63]

[1]Department of Plant Biology and Ecology, Faculty of Sciences and Technology, University of the Basque Country UPV/EHU, Leioa, Spain. [2]IKERBASQUE Basque Foundation for Science, Bilbao, Spain. [3]BC3 Basque Center for Climate Change, Leioa, Spain. [4]Department of Ecology and Evolution, Biophore, University of Lausanne, Lausanne, Switzerland. [5]Institute of Earth Surface Dynamics, Geopolis, University of Lausanne, Lausanne, Switzerland. [6]Institute of Ecology and Earth Sciences, University of Tartu, Tartu, Estonia. [7]Science and Art Faculty, Department of Biology, Lab of Ornithology, Burdur Mehmet Akif Ersoy University, Burdur, Turkey. [8]CIBIO-InBIO Laboratório Associado & Departamento de Biologia, Faculdade de Ciências do Porto, Campus de Vairão, Universidade do Porto, Vairão, Portugal. [9]BIOPOLIS Program in Genomics, Biodiversity and Land Planning, CIBIO, Campus de Vairão, Universidade do Porto, Vairão, Portugal. [10]EBM, Estação Biológica de Mértola, Mértola, Portugal. [11]Faculty of Agriculture, Forest Science and Natural Environment, Aristotle University of Thessaloniki, Thessaloniki, Greece. [12]Department of Biology, University of Copenhagen, Copenhagen, Denmark. [13]Institute of Nature Conservation, Polish Academy of Sciences, Kraków, Poland. [14]Faculty of Mathematics, Natural Sciences, and Information Technologies, University of Primorska, Koper, Slovenia. [15]Faculty of Environmental Protection, Velenje, Slovenia. [16]Department of Animal Science, University of Zagreb, Zagreb, Croatia. [17]Department of Biology and Ecology, Faculty of Sciences, University of Novi Sad, Novi Sad, Serbia. [18]Department of Wildlife, National Institute for Research and Development in Forestry 'Marin Dracea', Brasov, Romania. [19]Department of Silviculture, Faculty of Silviculture and Forest Engineering, Transilvania University of Brasov, Brasov, Romania. [20]Department of Medical Biochemistry and Microbiology, Uppsala University, Uppsala, Sweden. [21]Bavarian Office for Forest Genetics, Teisendorf, Germany. [22]Doñana Biological Station (EBD-CSIC), Seville, Spain. [23]Swiss Federal Research Institute WSL, Birmensdorf, Switzerland. [24]Center for Tree Science, Morton Arboretum, Lisle, IL, USA. [25]Department of Environmental Systems Sciences D-USYS, ETH Zürich, Zürich, Switzerland. [26]Copenhagen Zoo, Frederiksberg, Denmark. [27]Faculty of Mathematics, Natural Sciences and Information Technologies, Department of Biodiversity, University of Primorska, Koper, Slovenia. [28]Department of Veterinary Medicine, University of Sassari, Sassari, Italy. [29]Institute for Genetic Engineering and Biotechnology, University of Sarajevo, Sarajevo, Bosnia and Herzegovina. [30]Faculty of Forestry, Technical University in Zvolen, Zvolen, Slovak Republic. [31]Department of Forest Ecology, Faculty of Forestry and Wood Sciences, Czech University of Life Sciences, Prague, Czech Republic. [32]Norwegian Institute for Nature Research, Trondheim, Norway. [33]Department of Zoology, Division of Population Genetics, Stockholm University, Stockholm, Sweden. [34]Centre for Research in Anthropology, Lisbon, Portugal. [35]Institute for Nature Conservation and Forests, Lisbon, Portugal. [36]UCD School of Agriculture and Food Science, University College Dublin, Dublin, Ireland. [37]Research Institute for Nature and Forest, Geraardsbergen, Belgium. [38]Ecology, Evolution and Biodiversity Conservation, KU Leuven, Leuven, Belgium. [39]Institute of Silviculture, Department of Forest and Soil Sciences, University of Natural Resources and Life Sciences (BOKU), Vienna, Austria. [40]Department of Forest Nature Conservation, Forest Research Institute Baden-Württemberg, Freiburg, Germany. [41]Department of Biology, University of Iceland, Reykjavik, Iceland. [42]Department of Biology, Colorado State University, Fort Collins, CO, USA. [43]Faculty of Biological & Environmental Sciences, University of Helsinki, Helsinki, Finland. [44]Faculty of Biosciences and Aquaculture, Nord University, Bodø, Norway. [45]Israel Oceanographic and Limnological Research, National Institute of Oceanography, Haifa, Israel. [46]Institute of Vertebrate Biology, Czech Academy of Sciences, Brno, Czech Republic. [47]Genetic Resource Centre, Latvian State Forest Research Institute 'Silava', Salaspils, Latvia. [48]Wildlife Ecology and Management, University Freiburg, Freiburg, Germany. [49]Slovenian Forestry Institute, Ljubljana, Slovenia. [50]Swedish Species Information Centre, Swedish University of Agricultural Sciences, Uppsala, Sweden. [51]Institute of Biology/Geobotany and Botanical Garden, Martin Luther University Halle-Wittenberg, Halle (Saale), Germany. [52]German Centre for Integrative Biodiversity Research (iDiv) Halle-Jena-Leipzig, Leipzig, Germany. [53]Department of Forest Genetics, Physiology and Plantations, Forest Research Institute, Bulgarian Academy of Sciences, Sofia, Bulgaria. [54]Plant Ecology and Nature Conservation Group, Wageningen University, Wageningen, the Netherlands. [55]Forest Ecology Unit, Research and Innovation Centre, Fondazione Edmund Mach, San Michele all'Adige, Italy. [56]Natural History Museum Vienna, Vienna, Austria. [57]Department of Evolutionary Biology, University of Vienna, Vienna, Austria. [58]Department of Genetics, University of the Free State, Bloemfontein, South Africa. [59]School of Biosciences, Cardiff University, Cardiff, UK. [60]Department of Biochemistry, Genetics and Molecular Biology, University of Pretoria, Pretoria, South Africa. [61]These authors contributed equally: Peter B. Pearman, Olivier Broennimann. [62]These authors jointly supervised this work: Antoine Guisan, Michael Bruford. [63]Deceased: Michael Bruford. ✉e-mail: peter.b.pearman@gmail.com

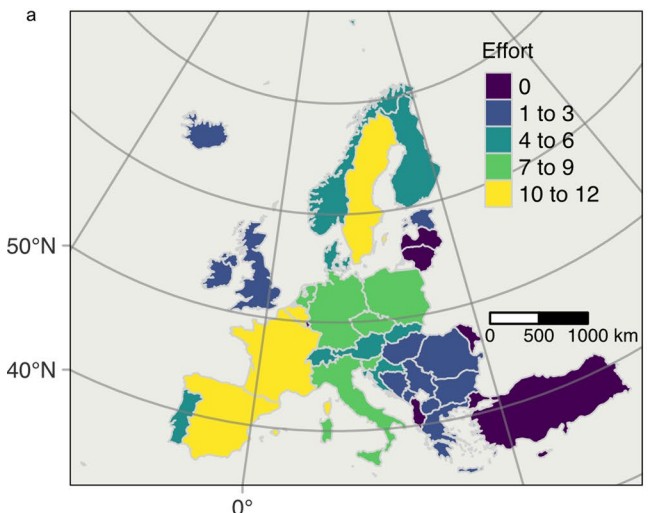

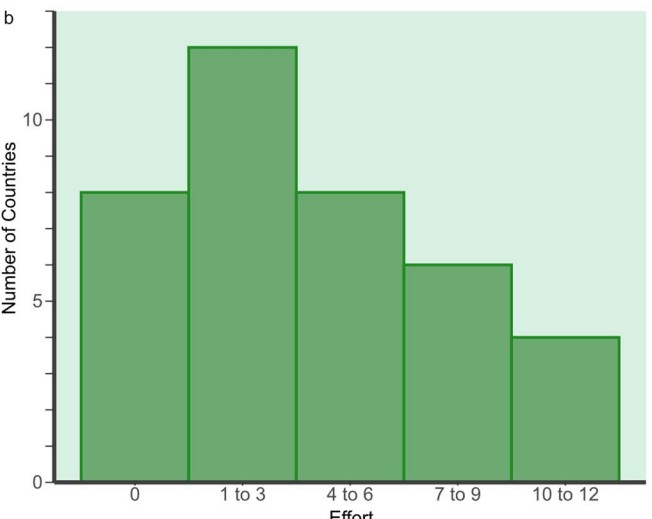

**Extended Data Fig. 1 | Documented programs to monitor population genetic diversity for conservation and management in COST member countries, as an indicator of genetic monitoring effort, GME, up to 31 December 2021. (a)** The geographic distribution of monitoring effort to countries, as a tally across all domestic and wild terrestrial and marine species indicates that countries with relatively high effort for monitoring are found in both northern and southern Europe. COST countries in southeastern Europe present generally low genetic monitoring effort. (**b**) The distribution of programs to countries shows that most countries have established six or fewer monitoring programs.

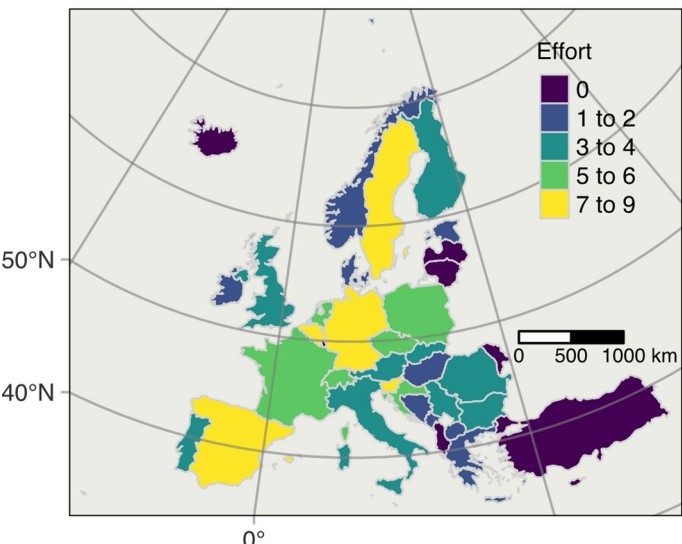

**Extended Data Fig. 2 | Genetic monitoring effort, GME, for terrestrial species.** The map shows for each COST Full-Member country the number of the monitoring projects up to 31 December 2021, The data include projects monitoring amphibians, birds, insects, carnivorans, other mammals and trees, but exclude programs/projects that monitored fish, marine species and domesticated/captive species.

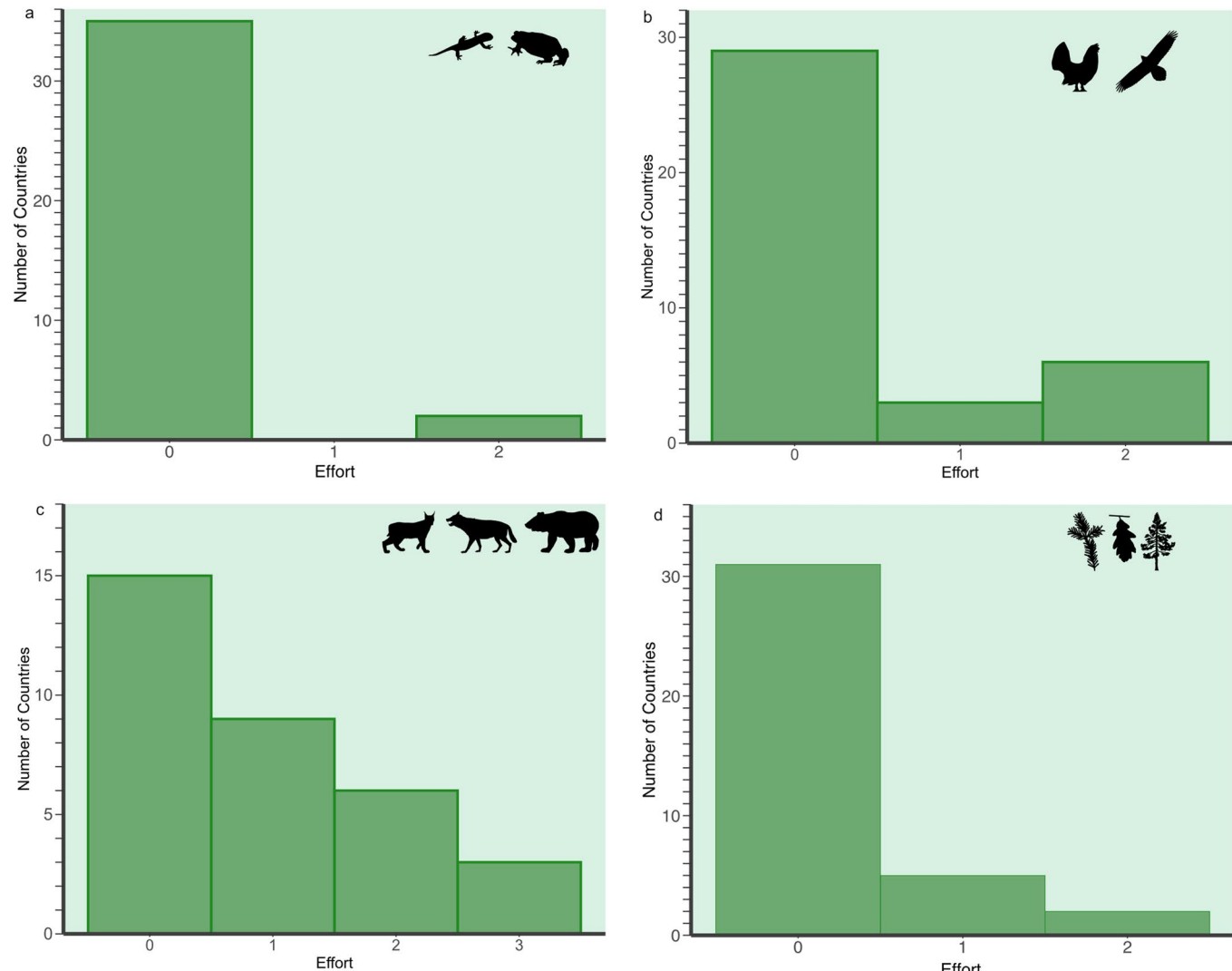

**Extended Data Fig. 3 | Frequency distribution of genetic monitoring effort among COST Full-Member Countries.** The data are for (**a**) amphibians, (**b**) birds, (**c**) large European carnivorans, and (**d**) forest trees. They represent all projects and programs using genetic data and reporting data on genetic diversity, from at least two time points, thus qualifying as Category II monitoring of population genetic diversity. Attribution for all silhouettes in this paper: https://www.phylopic.org/permalinks/ec1bdb9ee275ab1dc5a68e090a169a96c06c559e516989215bb5d881696db666.

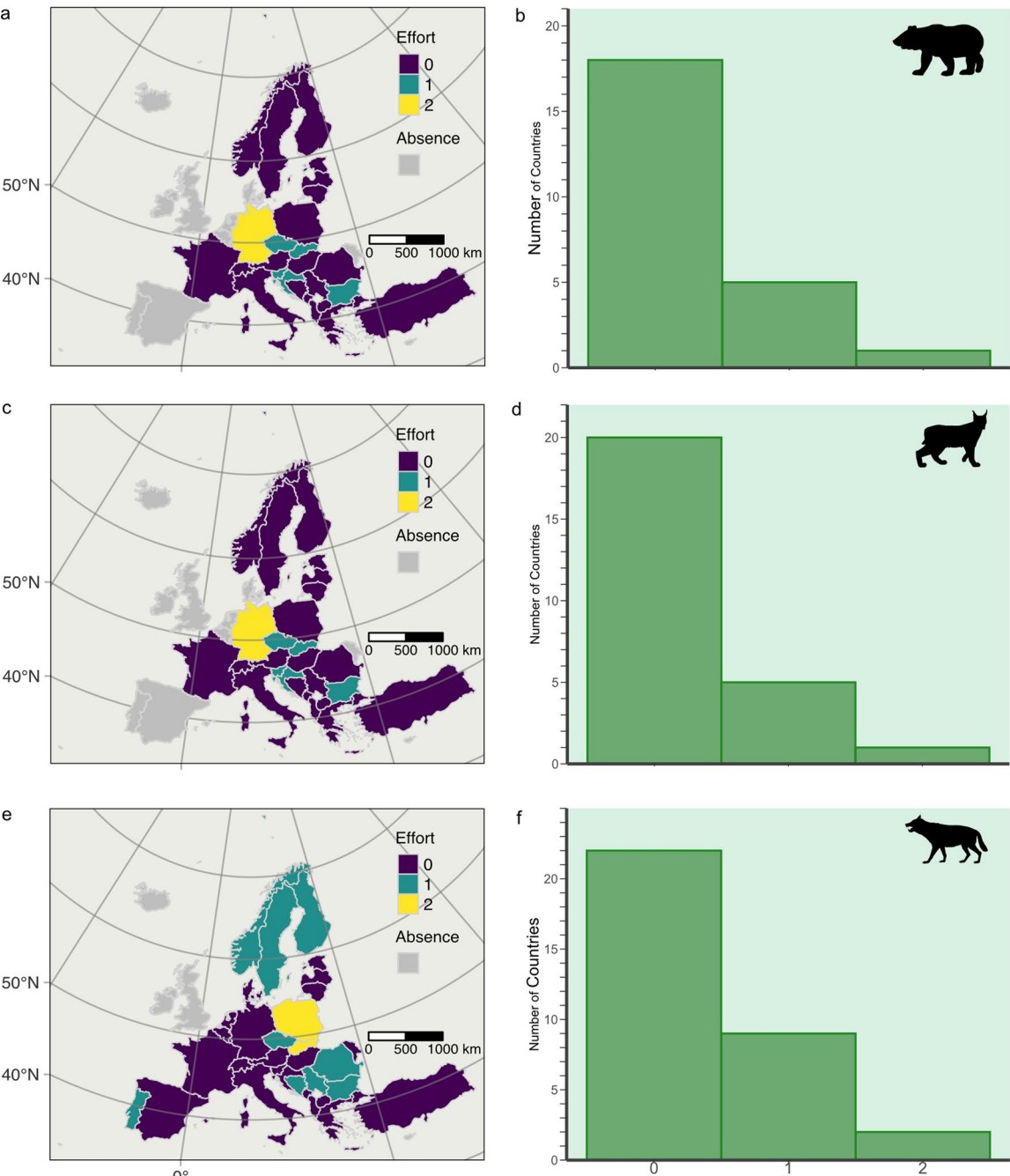

**Extended Data Fig. 4 | Distribution and frequency of monitoring programs for three carnivorans in COST Full-Member Countries.** The data are for (**a, b**) Eurasian brown bear, *Ursus arctos*, (**c, d**) Eurasian lynx, *Lynx lynx,* and (**e, f**) Eurasian wolf, *Canis lupus*. Tallies are of identifiably distinct monitoring projects and programs that report Category II monitoring of population genetic diversity.

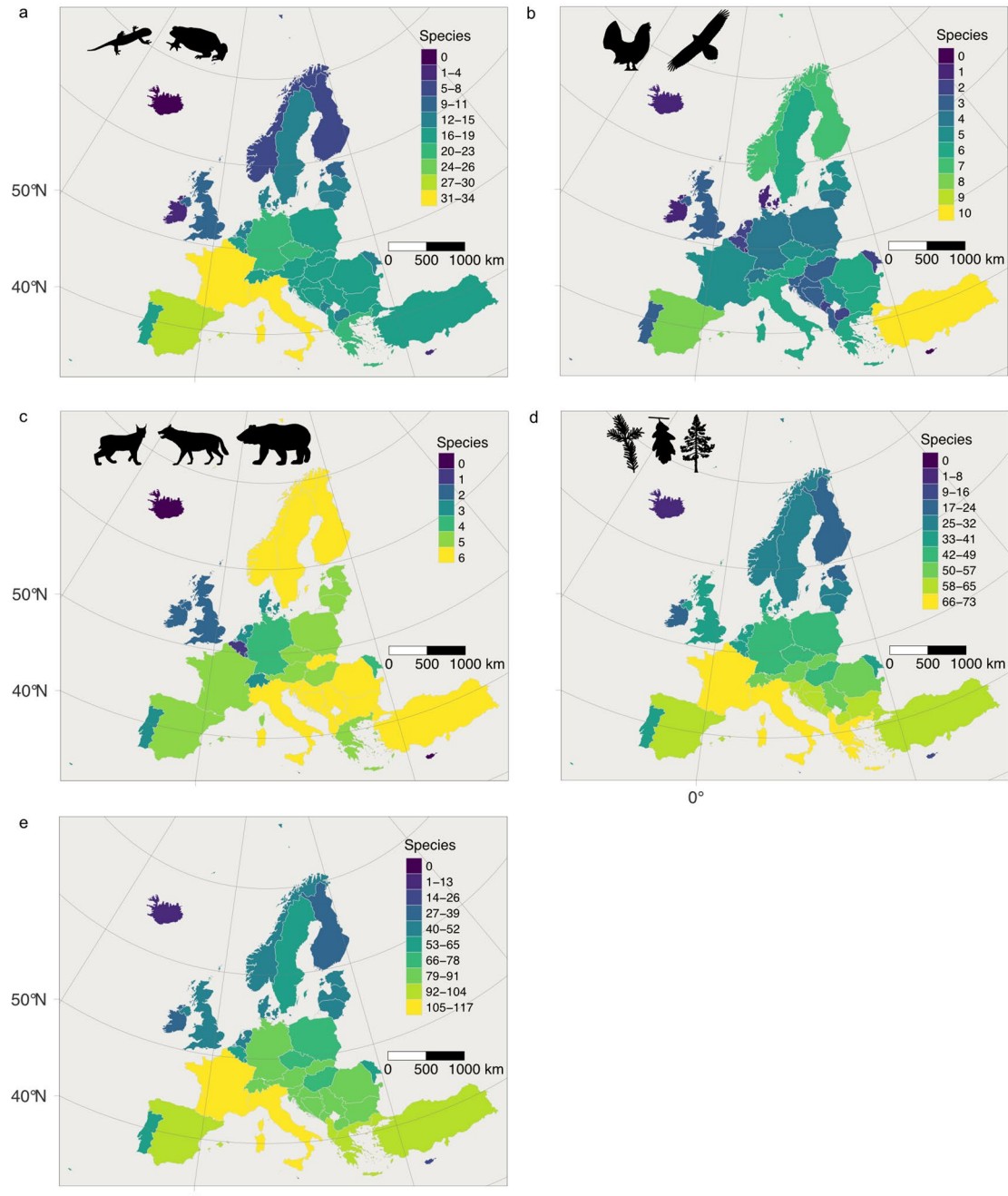

**Extended Data Fig. 5 | Number of species selected for investigation of patterns of niche marginality in COST Full-Member countries as of 31 January 2021.** The number of species with estimated range in each country is shown for (**a**) amphibians, (**b**) a selection of large birds, (**c**) a selection of large carnivores, (**d**) forest trees, and (**e**) all the study species together. The species were chosen for their current or potential future conservation or management interest. While some of the species have been the focus of monitoring of population genetic diversity, most have not, and this was not a requirement for their selection. See Extended Data Table 1 for species identities.

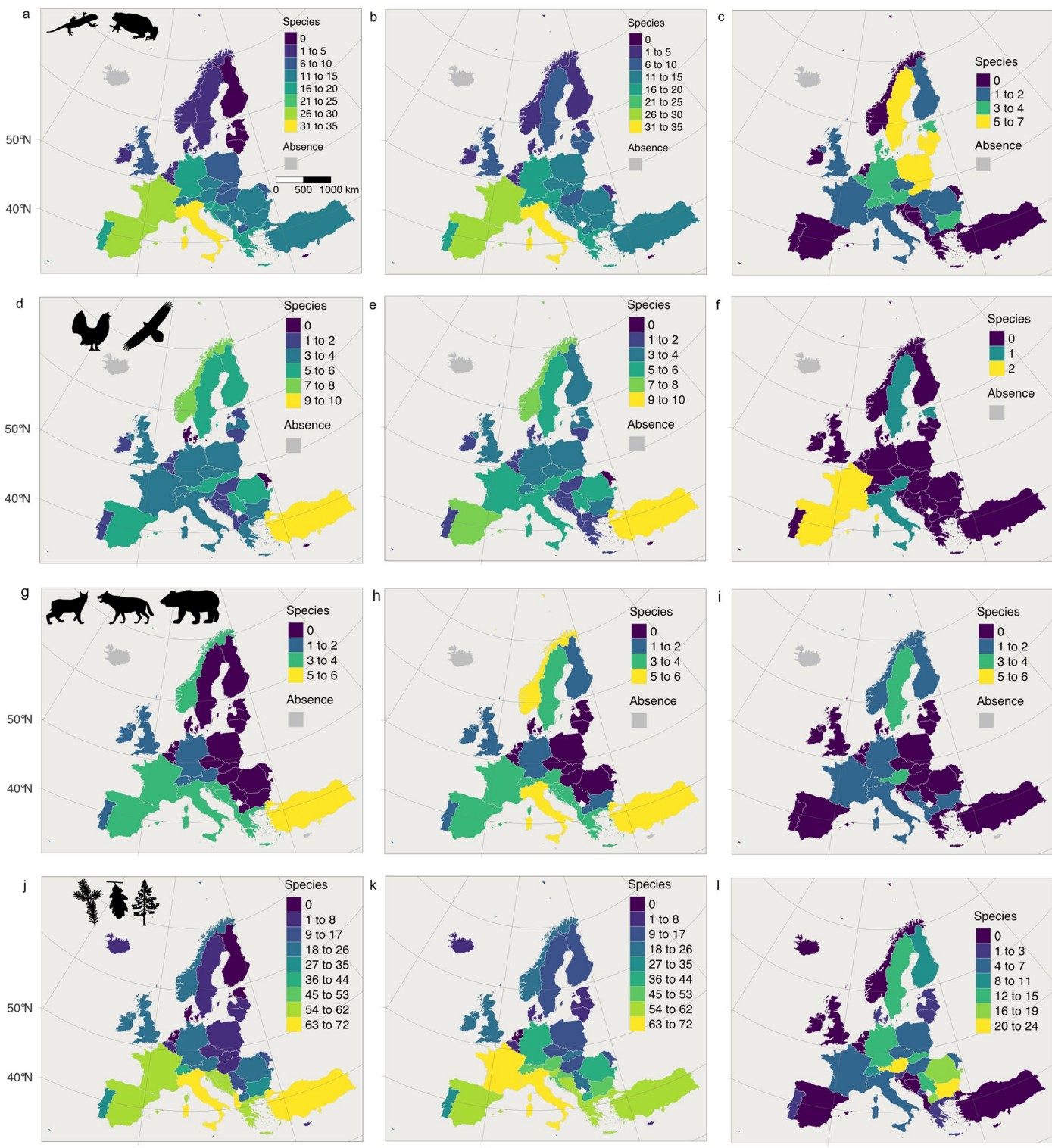

**Extended Data Fig. 6 | Number of species with trailing-edge, climatic niche margin conditions in each COST country.** Data shown for two times, current (1981-2010, left column) and future (2041-2070, center column), and number of species newly with trailing-edge marginal conditions in the future (right column). Values are generated with the UKESM1-0-LL GCM and SSP 5-8.5. The number of species are shown tallied by country for (**a, b, c**) 70 amphibian species, (**d, e, f**) a selection of 16 large bird species, (**g, h, i**) eight large carnivorans, and (**j, k, l**) 91 forest tree species. The patterns of changing joint niche marginality differ among species groups. An increasing number of populations newly at niche margins occurs in central and southeastern Europe in the study amphibians and trees (c, l, respectively). Populations of large carnivorans increasingly experience marginal niche conditions in Austria and Sweden (i). Populations of forest trees increasingly experience marginal niche conditions in southeastern Europe. Countries may show decreasing numbers of species with trailing-edge marginal conditions when populations experience conditions outside their niches in the future, throughout the country (for example, Turkey; j, k). See Appendix S7 in Supplementary Information for analogous data generated with other GCMs and SSPs. See Extended Data Table 1 for species identities.

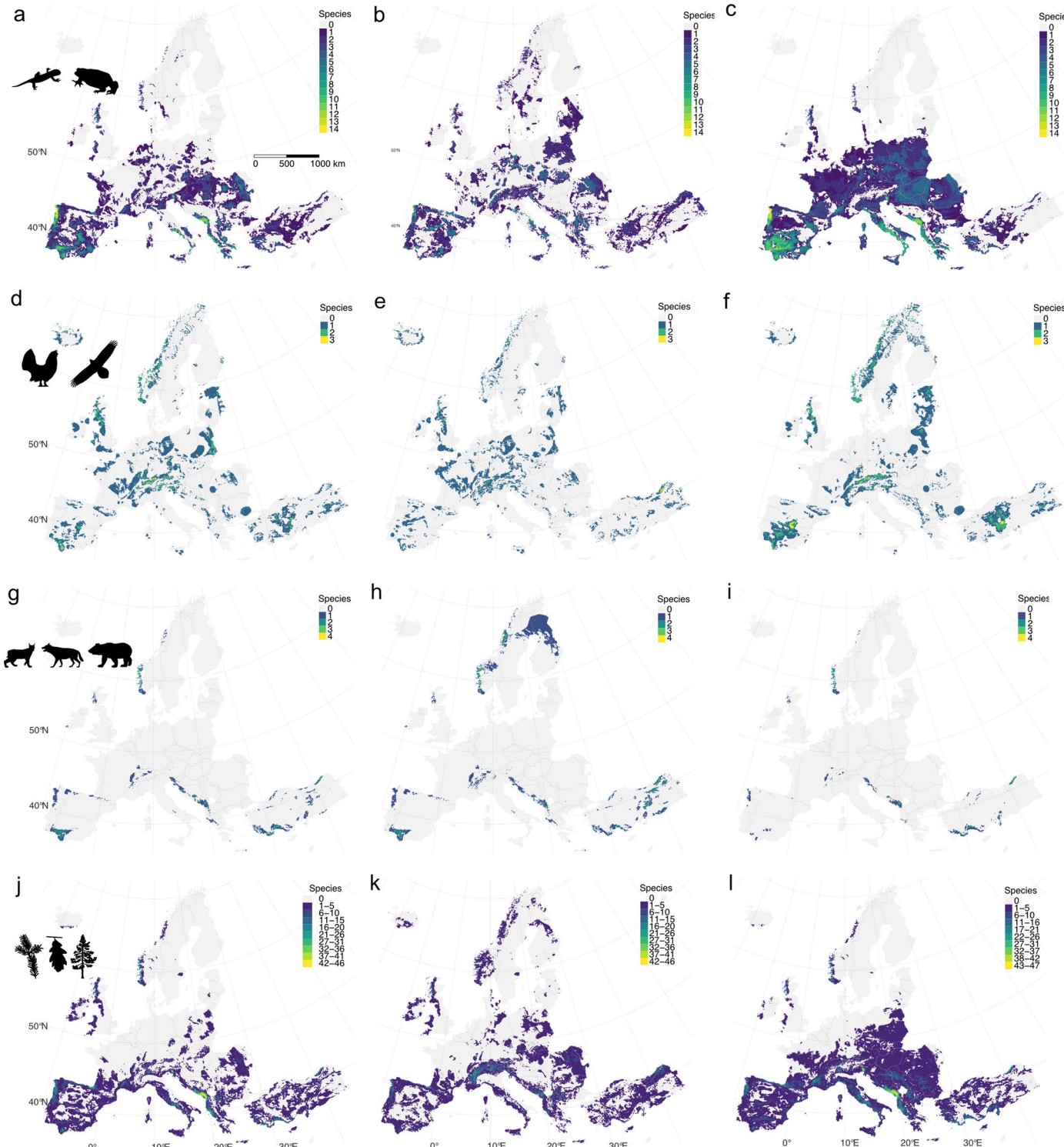

**Extended Data Fig. 7 | Changes in species joint niche marginality between current (1981-2010) and future (2031–2070).** Data are based on the UKESM1-0-LL Global Circulation Model and IPCC Shared Socioeconomic Pathway SSP 5-8.5. Numbers of species in four groups, with (left column) conditions currently within trailing edge niche marginality in COST full member countries; (middle column, dark green and yellow in Fig. 3) number of species with trailing niche marginality in the future time period; (right column, corresponding to red and orange in Fig. 3) number of species losing suitable climatic conditions between the current and future periods. Original data are at the level of 1 km² pixels, but here the data are aggregated to 10 km × 10 km pixels to improve visualization. The highest value within this 100 km² area is displayed. Values are for (**a, b, c**) amphibian species, (**d, e, f**) species of large birds, (**g, h, i**) several large carnivorans, and (**j, k, l**) a collection of forest tree species. Calculation of change in number of species with marginal niche conditions was based on species maps (Appendices S3-6, Supplemental Materials). See Extended Data Table 1 for species identities. See Methods for details on the calculation of climatic niche marginality for species and the filtering of pixels with a CORINE land cover layer.

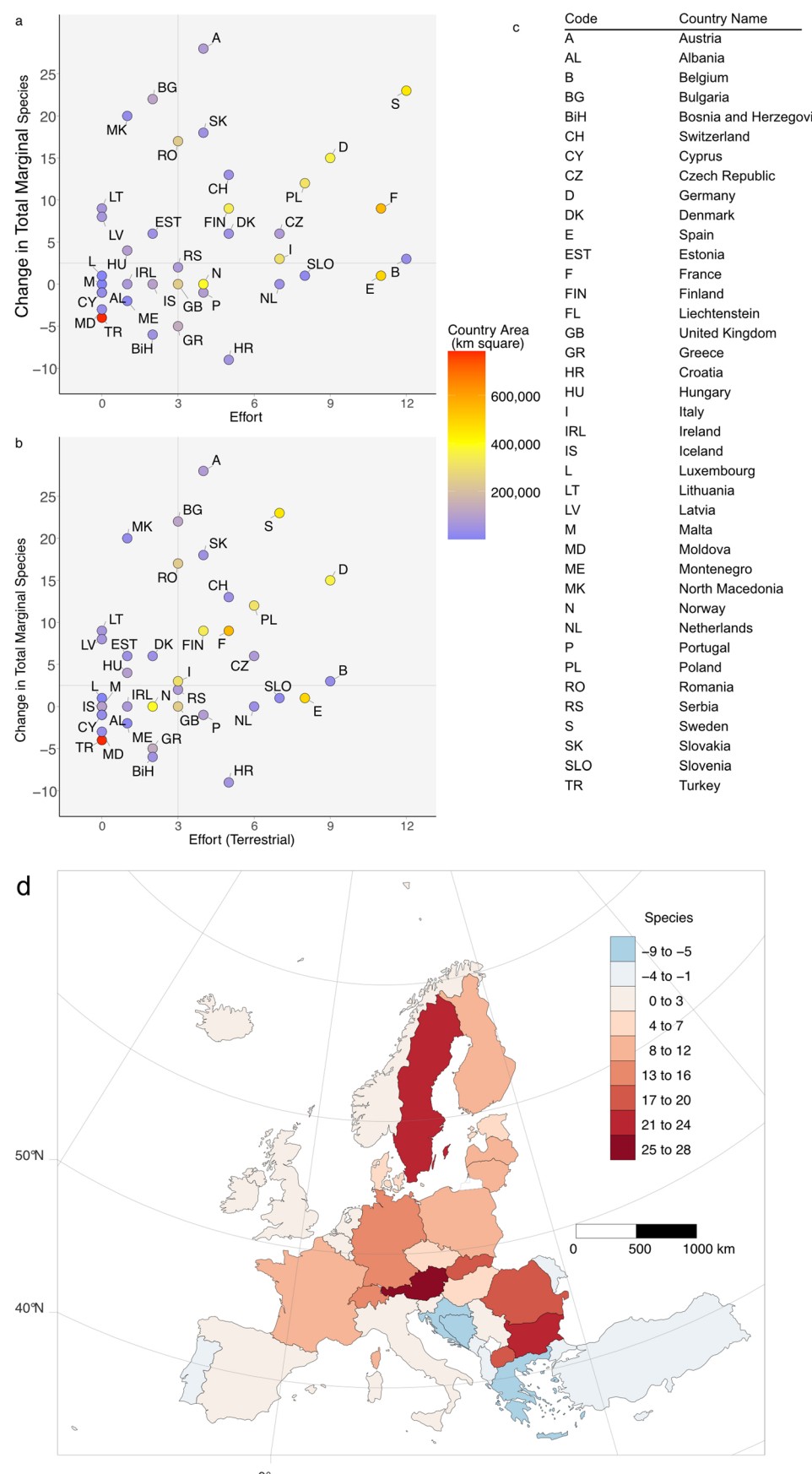

**Extended Data Fig. 8 | See next page for caption.**

**Extended Data Fig. 8 | Differences between current and future numbers of species that present trailing-edge niche marginality.** Data are based on the UKESM1-0-LL Global Circulation Model and IPCC Shared Socioeconomic Pathway SSP 5-8.5. The panels show (**a**) change in numbers of species with trailing-edge marginal conditions in each country versus monitoring effort, (**b**) change in numbers of trailing-edge marginal conditions versus terrestrial monitoring effort only, (**c**) country codes for 'a' and 'b', and (**d**) change in total number of species with trailing-edge marginal niche conditions between the current (1981-2010) and future (2031-2070) periods.

# Genetic Monitoring Validity

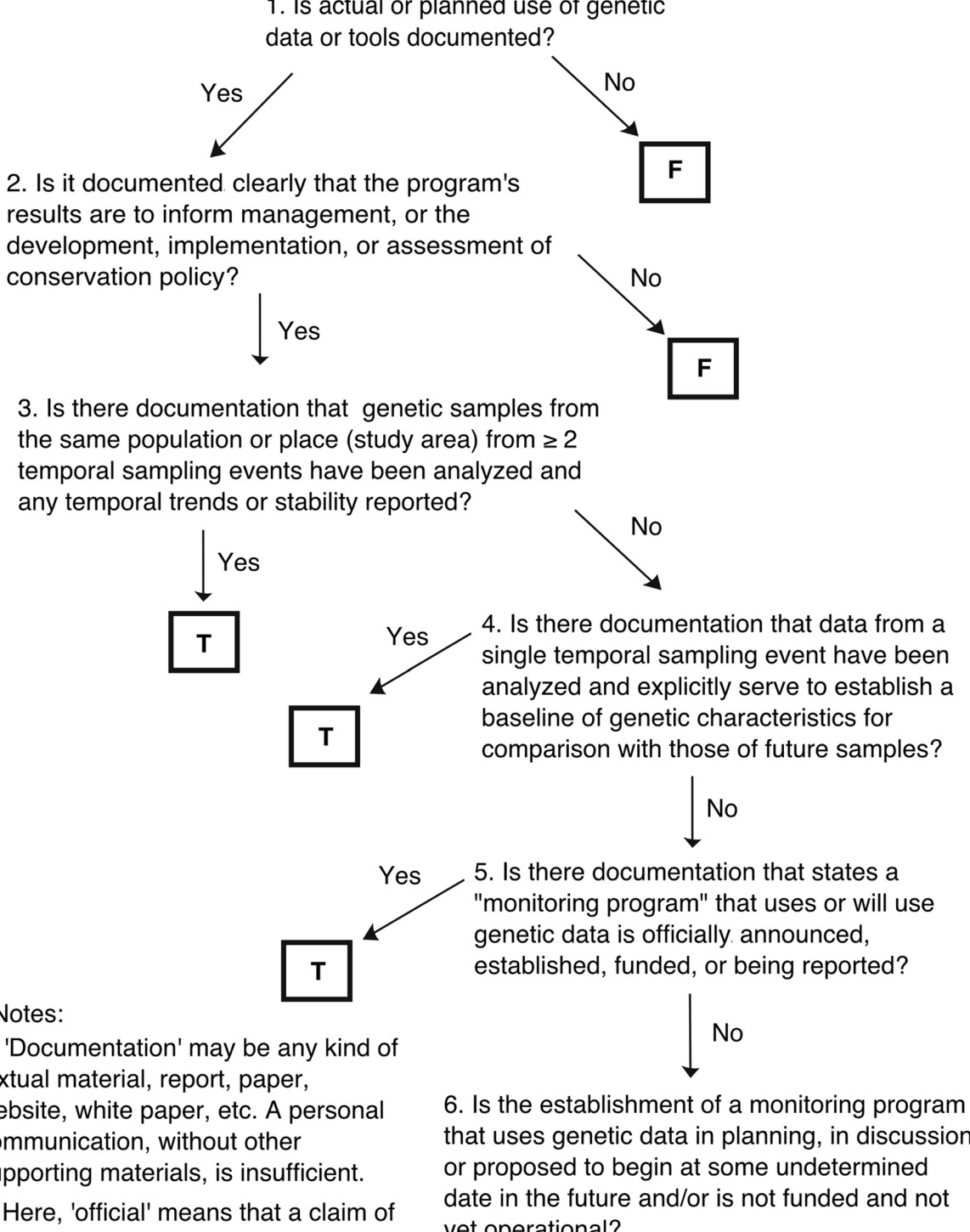

**1.** Is actual or planned use of genetic data or tools documented?

Yes / No → **F**

**2.** Is it documented clearly that the program's results are to inform management, or the development, implementation, or assessment of conservation policy?

Yes / No → **F**

**3.** Is there documentation that genetic samples from the same population or place (study area) from ≥ 2 temporal sampling events have been analyzed and any temporal trends or stability reported?

Yes → **T**

No

**4.** Is there documentation that data from a single temporal sampling event have been analyzed and explicitly serve to establish a baseline of genetic characteristics for comparison with those of future samples?

Yes → **T**

No

**5.** Is there documentation that states a "monitoring program" that uses or will use genetic data is officially announced, established, funded, or being reported?

Yes → **T**

No

Notes:

1. 'Documentation' may be any kind of textual material, report, paper, website, white paper, etc. A personal communication, without other supporting materials, is insufficient.

2. Here, 'official' means that a claim of the existence of a "monitoring program" with the stated characteristics has been publicly released in some way by a governmental or non-governmental organization.

**6.** Is the establishment of a monitoring program that uses genetic data in planning, in discussion, or proposed to begin at some undetermined date in the future and/or is not funded and not yet operational?

Yes / No → **F**

**Extended Data Fig. 9 | Decision diagram.** A flow chart for supporting decisions on the validity of projects as constituting valid Category II genetic monitoring given a wide range of potential documentation, originating in government reports, web documents, and the peer-reviewed literature.

**Extended Data Table 1 | Species of current or potential future conservation or management interest and used in this study for examining niche marginality**

| Group | subgroup | GBIKE_name | GBIF_name | IUCN_name | english_name | remark |
|---|---|---|---|---|---|---|
| amphibians | Anura | Alytes cisternasii | Alytes cisternasii | Alytes cisternasii | Iberian midwife toad | |
| amphibians | Anura | Alytes dickhilleni | Alytes dickhilleni | Alytes dickhilleni | Betic Midwife Toad | |
| amphibians | Anura | Alytes obstetricans | Alytes obstetricans | Alytes obstetricans | Midwife toad | |
| amphibians | Anura | Bombina bombina | Bombina bombina | Bombina bombina | European fire-bellied toad | |
| amphibians | Anura | Bombina pachypus | Bombina pachypus | Bombina pachypus | Appenine Yellow-bellied Toad | |
| amphibians | Anura | Bombina variegata | Bombina variegata | Bombina variegata | Yellow-bellied toad | |
| amphibians | Anura | Bufo bufo | Bufo bufo | Bufo bufo | Common European Toad | |
| amphibians | Anura | Discoglossus galganoi | Discoglossus galganoi | Discoglossus galganoi | Iberian painted frog | |
| amphibians | Anura | Discoglossus montalentii | Discoglossus montalentii | Discoglossus montalentii | Corsica Painted Frog | |
| amphibians | Anura | Discoglossus pictus | Discoglossus pictus | Discoglossus pictus | Painted frog | |
| amphibians | Anura | Discoglossus sardus | Discoglossus sardus | Discoglossus sardus | Sardinia painted frog | |
| amphibians | Anura | Epidalea calamita | Epidalea calamita | Epidalea calamita | Natterjack toad | |
| amphibians | Anura | Hyla arborea | Hyla arborea | Hyla arborea | European tree frog | |
| amphibians | Anura | Hyla intermedia | Hyla intermedia | Hyla intermedia | Italian Tree Frog | |
| amphibians | Anura | Hyla meridionalis | Hyla meridionalis | Hyla meridionalis | Mediterranean Treefrog | |
| amphibians | Anura | Hyla sarda | Hyla sarda | Hyla sarda | Sardinian Tree Frog | |
| amphibians | Anura | Hyla savignyi | Hyla savignyi | Hyla savignyi | Savigny's treefrog | |
| amphibians | Anura | Pelobates cultripes | Pelobates cultripes | Pelobates cultripes | western spadefoot | |
| amphibians | Anura | Pelobates fuscus | Pelobates fuscus | Pelobates fuscus | Common Eurasian spadefoot toad | |
| amphibians | Anura | Pelobates syriacus | Pelobates syriacus | Pelobates syriacus | Syrian spadefoot toad | |
| amphibians | Anura | Pelodytes ibericus | Pelodytes ibericus | Pelodytes ibericus | Iberian spadefoot toad | |
| amphibians | Anura | Pelodytes punctatus | Pelodytes punctatus | Pelodytes punctatus | Parsley frog | |
| amphibians | Anura | Pelophylax bedriagae | Pelophylax bedriagae | Pelophylax bedriagae | Bedriaga's Frog | |
| amphibians | Anura | Pelophylax bergeri | Pelophylax bergeri | Pelophylax bergeri | Italian Pool Frog | |
| amphibians | Anura | Pelophylax cerigensis | Pelophylax cerigensis | Pelophylax cerigensis | Karpathos Frog | |
| amphibians | Anura | Pelophylax cretensis | Pelophylax cretensis | Pelophylax cretensis | Cretan Frog | |
| amphibians | Anura | Pelophylax epeiroticus | Pelophylax epeiroticus | Pelophylax epeiroticus | Epirus Water Frog | |
| amphibians | Anura | Pelophylax kurtmuelleri | Pelophylax kurtmuelleri | Pelophylax kurtmuelleri | Balkan Water Frog | |
| amphibians | Anura | Pelophylax lessonae | Pelophylax lessonae | Pelophylax lessonae | Pool Frog | |
| amphibians | Anura | Pelophylax perezi | Pelophylax perezi | Pelophylax perezi | Perez's Frog | |
| amphibians | Anura | Pelophylax ridibundus | Pelophylax ridibundus | Pelophylax ridibundus | Marsh frog | |
| amphibians | Anura | Pelophylax shqipericus | Pelophylax shqipericus | Pelophylax shqipericus | Albanian Water Frog | |
| amphibians | Anura | Pseudepidalea balearica | Bufotes balearicus | Bufotes balearicus | Balearic green toad | |
| amphibians | Anura | Pseudepidalea sicula | Bufotes siculus | Bufotes siculus | African Green Toad | Bufotes boulengeri subsp. Siculus |
| amphibians | Anura | Pseudepidalea variabilis | Bufotes variabilis | Bufotes variabilis | Varying Toad | |
| amphibians | Anura | Pseudepidalea viridis | Bufotes viridis | Bufotes viridis | Green Toad | |
| amphibians | Anura | Rana arvalis | Rana arvalis | Rana arvalis | Moor frog | |
| amphibians | Anura | Rana dalmatina | Rana dalmatina | Rana dalmatina | Agile frog | |
| amphibians | Anura | Rana graeca | Rana graeca | Rana graeca | Stream frog | |
| amphibians | Anura | Rana iberica | Rana iberica | Rana iberica | Iberian frog | |
| amphibians | Anura | Rana italica | Rana italica | Rana italica | Italian stream frog | |
| amphibians | Anura | Rana latastei | Rana latastei | Rana latastei | Italian agile frog | |
| amphibians | Anura | Rana pyrenaica | Rana pyrenaica | Rana pyrenaica | Pyrenean Frog | |
| amphibians | Anura | Rana temporaria | Rana temporaria | Rana temporaria | Common frog | |
| amphibians | Caudata | Calotriton arnoldi | Calotriton arnoldi | Calotriton arnoldi | Montseny brook newt | |
| amphibians | Caudata | Calotriton asper | Calotriton asper | Calotriton asper | Pyrenean Brook Salamander | |
| amphibians | Caudata | Chioglossa lusitanica | Chioglossa lusitanica | Chioglossa lusitanica | Gold-striped salamander | |
| amphibians | Caudata | Euproctus montanus | Euproctus montanus | Euproctus montanus | Corsican mountain newt | |
| amphibians | Caudata | Euproctus platycephalus | Euproctus platycephalus | Euproctus platycephalus | Sardinian mountain newt | |
| amphibians | Caudata | Lissotriton boscai | Lissotriton boscai | Lissotriton boscai | Bosca's Newt | |
| amphibians | Caudata | Lissotriton helveticus | Lissotriton helveticus | Lissotriton helveticus | Palmate Newt | |
| amphibians | Caudata | Lissotriton italicus | Lissotriton italicus | Lissotriton italicus | Italian Newt | |
| amphibians | Caudata | Lissotriton montandoni | Lissotriton montandoni | Lissotriton montandoni | Carpathian Newt | |
| amphibians | Caudata | Lissotriton vulgaris | Lissotriton vulgaris | Lissotriton vulgaris | Smooth Newt | |
| amphibians | Caudata | Lyciasalamandra helverseni | Lyciasalamandra helverseni | Lyciasalamandra helverseni | Karpathos salamander | |
| amphibians | Caudata | Lyciasalamandra luschani | Lyciasalamandra luschani | Lyciasalamandra luschani | Luschan's salamander | |
| amphibians | Caudata | Mesotriton alpestris | Ichthyosaura alpestris | Ichthyosaura alpestris | Alpine Newt | |
| amphibians | Caudata | Pleurodeles waltl | Pleurodeles waltl | Pleurodeles waltl | Iberian ribbed newt | |
| amphibians | Caudata | Salamandra atra | Salamandra atra | Salamandra atra | Alpine salamander | |
| amphibians | Caudata | Salamandra corsica | Salamandra corsica | Salamandra corsica | Corsican Fire Salamander | |
| amphibians | Caudata | Salamandra lanzai | Salamandra lanzai | Salamandra lanzai | Lanza's Alpine salamander | |
| amphibians | Caudata | Salamandra salamandra | Salamandra salamandra | Salamandra salamandra | Fire salamander | |
| amphibians | Caudata | Salamandrina perspicillata | Salamandrina perspicillata | Salamandrina perspicillata | Northern spectacled salamander | |
| amphibians | Caudata | Salamandrina terdigitata | Salamandrina terdigitata | Salamandrina terdigitata | Southern spectacled salamander | |
| amphibians | Caudata | Triturus carnifex | Triturus carnifex | Triturus carnifex | Italian crested newt | |
| amphibians | Caudata | Triturus cristatus | Triturus cristatus | Triturus cristatus | Crested newt | |
| amphibians | Caudata | Triturus dobrogicus | Triturus dobrogicus | Triturus dobrogicus | Danube Crested Newt | |
| amphibians | Caudata | Triturus karelinii | Triturus karelinii | Triturus karelinii | Southern Crested Newt | |
| amphibians | Caudata | Triturus marmoratus | Triturus marmoratus | Triturus marmoratus | Marbled newt | |
| amphibians | Caudata | Triturus pygmaeus | Triturus pygmaeus | Triturus pygmaeus | Southern Marbled Newt | |
| bigbirds | Accipitridae | Aquila adalberti | Aquila adalberti | Aquila adalberti | Spanish Imperial Eagle | |
| bigbirds | Accipitridae | Aquila heliaca | Aquila heliaca | Aquila heliaca | Eastern Imperial Eagle | |
| bigbirds | Accipitridae | Clanga clanga | Clanga clanga | Clanga clanga | Greater Spotted Eagle | |
| bigbirds | Accipitridae | Gypaetus barbatus | Gypaetus barbatus | Gypaetus barbatus | Bearded Vulture | |
| bigbirds | Accipitridae | Neophron percnopterus | Neophron percnopterus | Neophron percnopterus | Egyptian Vulture | |
| bigbirds | Anatidae | Anser erythropus | Anser erythropus | Anser erythropus | Lesser White-fronted Goose | |
| bigbirds | Anatidae | Aythya ferina | Aythya ferina | Aythya ferina | Common Pochard | |
| bigbirds | Anatidae | Clangula hyemalis | Clangula hyemalis | Clangula hyemalis | Long-tailed Duck | |
| bigbirds | Anatidae | Marmaronetta angustirostris | Marmaronetta angustirostris | Marmaronetta angustirostris | Marbled Teal | |
| bigbirds | Anatidae | Melanitta fusca | Melanitta fusca | Melanitta fusca | Velvet Scoter | |
| bigbirds | Anatidae | Oxyura leucocephala | Oxyura leucocephala | Oxyura leucocephala | White-headed Duck | |
| bigbirds | Anatidae | Polysticta stelleri | Polysticta stelleri | Polysticta stelleri | Steller's Eider | |
| bigbirds | Gallidae | Lyrurus mlokosiewiczi | Lyrurus mlokosiewiczi | Lyrurus mlokosiewiczi | Caucasian Grouse | |
| bigbirds | Gallidae | Lyrurus tetrix | Lyrurus tetrix | Lyrurus tetrix | Black Grouse | |
| bigbirds | Gallidae | Tetrao urogallus | Tetrao urogallus | Tetrao urogallus | Western Capercaillie | |
| bigbirds | Otididae | Otis tarda | Otis tarda | Otis tarda | Great Bustard | |
| carnivores | Canidae | Canis aureus | Canis aureus | Canis aureus | Golden Jackal | |
| carnivores | Canidae | Canis lupus | Canis lupus | Canis lupus | Wolf | |
| carnivores | Mustelidae | Gulo gulo | Gulo gulo | Gulo gulo | Wolverine | |
| carnivores | Mustelidae | Lutra lutra | Lutra lutra | Lutra lutra | Eurasian Otter | |
| carnivores | Felidae | Lynx lynx | Lynx lynx | Lynx lynx | Eurasian Lynx | |
| carnivores | Felidae | Lynx pardinus | Lynx pardinus | Lynx pardinus | Iberian Lynx | |
| carnivores | Mustelidae | Meles meles | Meles meles | Meles meles | European Badger | |
| carnivores | Ursidae | Ursus arctos | Ursus arctos | Ursus arctos | Eurasian brown Bear | |
| trees | Magnoliopsida | Acer campestre | | | Field maple | |
| trees | Magnoliopsida | Acer monspessulanum | | | Montpelier maple | |
| trees | Magnoliopsida | Acer platanoides | | | Norway maple | |
| trees | Magnoliopsida | Acer pseudoplatanus | | | Sycamore | |
| trees | Magnoliopsida | Aesculus hippocastanum | | | horse chestnut | |
| trees | Magnoliopsida | Alnus cordata | | | Italian alder | |
| trees | Magnoliopsida | Alnus glutinosa | | | Black alder | |
| trees | Magnoliopsida | Alnus incana | | | Grey alder | ssp. Incana only |
| trees | Magnoliopsida | Alnus viridis | | | Green alder | ssp. Viridis only |
| trees | Magnoliopsida | Arbutus unedo | | | strawberry tree | |
| trees | Magnoliopsida | Betula pendula | | | Silver birch | |
| trees | Magnoliopsida | Betula pubescens | | | Downy birch | |
| trees | Magnoliopsida | Buxus sempervirens | | | common box | |
| trees | Magnoliopsida | Carpinus betulus | | | European hornbeam | |
| trees | Magnoliopsida | Carpinus orientalis | | | Oriental hornbeam | |
| trees | Magnoliopsida | Castanea sativa | | | Chestnut | |
| trees | Magnoliopsida | Celtis australis | | | European nettle tree | |
| trees | Magnoliopsida | Cornus mas | | | Cornelian cherry | |
| trees | Magnoliopsida | Cornus sanguinea | | | Common dogwood | |
| trees | Magnoliopsida | Corylus avellana | | | Common hazel | |
| trees | Magnoliopsida | Euonymus europaeus | | | common spindle | |
| trees | Magnoliopsida | Fagus sylvatica | | | European beech | |
| trees | Magnoliopsida | Frangula alnus | | | Glossy buckthorn | |
| trees | Magnoliopsida | Fraxinus angustifolia | | | Narrow-leaved ash | |
| trees | Magnoliopsida | Fraxinus excelsior | | | Common ash | |
| trees | Magnoliopsida | Fraxinus ornus | | | manna ash | |
| trees | Magnoliopsida | Ilex aquifolium | | | Common holly | |
| trees | Magnoliopsida | Juglans regia | | | Common walnut | |
| trees | Magnoliopsida | Juniperus phoenicea | | | Phoenican juniper | |
| trees | Magnoliopsida | Juniperus thurifera | | | Spanish juniper | |
| trees | Magnoliopsida | Liquidambar orientalis | | | Oriental sweet gum | |
| trees | Magnoliopsida | Olea europaea | | | European olive | |
| trees | Magnoliopsida | Ostrya carpinifolia | | | Hop hornbeam | |
| trees | Magnoliopsida | Platanus orientalis | | | Oriental plane | |
| trees | Magnoliopsida | Populus alba | | | White poplar | |
| trees | Magnoliopsida | Populus nigra | | | European black poplar | |
| trees | Magnoliopsida | Populus tremula | | | Eurasian aspen | |
| trees | Magnoliopsida | Prunus avium | | | Wild cherry | |
| trees | Magnoliopsida | Prunus padus | | | Bird cherry | |
| trees | Magnoliopsida | Prunus spinosa | | | Blackthorn | |
| trees | Magnoliopsida | Quercus cerris | | | Turkey oak | |
| trees | Magnoliopsida | Quercus coccifera | | | Kermes oak | |
| trees | Magnoliopsida | Quercus faginea | | | Portuguese oak | |
| trees | Magnoliopsida | Quercus frainetto | | | Hungarian oak | |
| trees | Magnoliopsida | Quercus ilex | | | Holm oak | |
| trees | Magnoliopsida | Quercus petraea | | | Sessile oak | |
| trees | Magnoliopsida | Quercus pubescens | | | Pubescent oak | |
| trees | Magnoliopsida | Quercus pyrenaica | | | Pyrenean oak | |
| trees | Magnoliopsida | Quercus robur | | | Pedunculate oak | |
| trees | Magnoliopsida | Quercus suber | | | Cork oak | |
| trees | Magnoliopsida | Quercus trojana | | | Macedonian oak | |
| trees | Magnoliopsida | Salix alba | | | White willow | |
| trees | Magnoliopsida | Salix caprea | | | Goat willow | |
| trees | Magnoliopsida | Sambucus nigra | | | black elder | |
| trees | Magnoliopsida | Sorbus aria | | | common whitebeam | |
| trees | Magnoliopsida | Sorbus aucuparia | | | European mountain ash | |
| trees | Magnoliopsida | Sorbus domestica | | | Service tree | |
| trees | Magnoliopsida | Sorbus torminalis | | | Wild service tree | |
| trees | Magnoliopsida | Tilia cordata | | | Small-leaved lime | |
| trees | Magnoliopsida | Tilia platyphyllos | | | Large-leaved lime | |
| trees | Magnoliopsida | Tilia tomentosa | | | Silver lime | |
| trees | Magnoliopsida | Ulmus glabra | | | Wych elm | |
| trees | Magnoliopsida | Ulmus laevis | | | European white elm | |
| trees | Magnoliopsida | Ulmus minor | | | Field elm | |
| trees | Pinopsida | Abies alba | | | Silver fir | |
| trees | Pinopsida | Abies borisii-regis | | | King Boris fir | dash in the name |
| trees | Pinopsida | Abies cephalonica | | | Grecian fir | |
| trees | Pinopsida | Abies cilicica | | | Cilician fir | |
| trees | Pinopsida | Abies nebrodensis | | | Sicilian fir | |
| trees | Pinopsida | Abies nordmanniana | | | Caucasian fir | |
| trees | Pinopsida | Abies numidica | | | Algerian fir | |
| trees | Pinopsida | Abies pinsapo | | | Spanish fir | |
| trees | Pinopsida | Cedrus libani | | | Cedar of Lebanon | |
| trees | Pinopsida | Cupressus sempervirens | | | Italian cypress | |
| trees | Pinopsida | Juniperus communis | | | Common juniper | |
| trees | Pinopsida | Juniperus excelsa | | | Greek juniper | |
| trees | Pinopsida | Juniperus oxycedrus | | | Prickly juniper | |
| trees | Pinopsida | Larix decidua | | | European larch | |
| trees | Pinopsida | Picea abies | | | Norway spruce | |
| trees | Pinopsida | Picea omorika | | | Serbian spruce | |
| trees | Pinopsida | Pinus brutia | | | Brutia pine | |
| trees | Pinopsida | Pinus cembra | | | Swiss stone pine | |
| trees | Pinopsida | Pinus halepensis | | | Aleppo pine | |
| trees | Pinopsida | Pinus heldreichii | | | Bosnian pine | |
| trees | Pinopsida | Pinus mugo | | | Mountain pine | |
| trees | Pinopsida | Pinus nigra | | | European black pine | |
| trees | Pinopsida | Pinus peuce | | | Macedonian pine | |
| trees | Pinopsida | Pinus pinaster | | | Maritime pine | |
| trees | Pinopsida | Pinus pinea | | | Stone pine | |
| trees | Pinopsida | Pinus sylvestris | | | Scots pine | |
| trees | Pinopsida | Taxus baccata | | | Common yew | |

# Reporting Summary

## Statistics

For all statistical analyses, confirm that the following items are present in the figure legend, table legend, main text, or Methods section.

| n/a | Confirmed | |
|---|---|---|
| ☐ | ☒ | The exact sample size (*n*) for each experimental group/condition, given as a discrete number and unit of measurement |
| ☒ | ☐ | A statement on whether measurements were taken from distinct samples or whether the same sample was measured repeatedly |
| ☐ | ☒ | The statistical test(s) used AND whether they are one- or two-sided<br>*Only common tests should be described solely by name; describe more complex techniques in the Methods section.* |
| ☐ | ☒ | A description of all covariates tested |
| ☐ | ☒ | A description of any assumptions or corrections, such as tests of normality and adjustment for multiple comparisons |
| ☐ | ☒ | A full description of the statistical parameters including central tendency (e.g. means) or other basic estimates (e.g. regression coefficient) AND variation (e.g. standard deviation) or associated estimates of uncertainty (e.g. confidence intervals) |
| ☐ | ☒ | For null hypothesis testing, the test statistic (e.g. *F*, *t*, *r*) with confidence intervals, effect sizes, degrees of freedom and *P* value noted<br>*Give P values as exact values whenever suitable.* |
| ☒ | ☐ | For Bayesian analysis, information on the choice of priors and Markov chain Monte Carlo settings |
| ☒ | ☐ | For hierarchical and complex designs, identification of the appropriate level for tests and full reporting of outcomes |
| ☒ | ☐ | Estimates of effect sizes (e.g. Cohen's *d*, Pearson's *r*), indicating how they were calculated |

*Our web collection on statistics for biologists contains articles on many of the points above.*

## Software and code

Policy information about availability of computer code

| Data collection | Not complicated.  Only a Google sheet was used to collect candidate monitoring cases. Climate data, species range polygons, and occurrence data were acquired from public sources. |
|---|---|
| Data analysis | Analyses are described in detail in the text.  Methods of spatial statistics and control of spatial autocorrelation are described in the on-line methods section.  Statistical code and results are provided in Appendix S2. Raw data and code for marginality calculations and mapping is provided in a Zenodo repository. |

For manuscripts utilizing custom algorithms or software that are central to the research but not yet described in published literature, software must be made available to editors and reviewers. We strongly encourage code deposition in a community repository (e.g. GitHub). See the Nature Portfolio guidelines for submitting code & software for further information.

## Data

Policy information about availability of data

All manuscripts must include a data availability statement. This statement should provide the following information, where applicable:

- Accession codes, unique identifiers, or web links for publicly available datasets
- A description of any restrictions on data availability
- For clinical datasets or third party data, please ensure that the statement adheres to our policy

Original candidate monitoring project submissions are included in Appendix S11 and a DOI for the data set at a repository is provided in a data availability

statement.

## Research involving human participants, their data, or biological material

Policy information about studies with underline{human participants or human data}. See also policy information about underline{sex, gender (identity/presentation), and sexual orientation} and underline{race, ethnicity and racism}.

| | |
|---|---|
| Reporting on sex and gender | n/a |
| Reporting on race, ethnicity, or other socially relevant groupings | n/a |
| Population characteristics | n/a |
| Recruitment | n/a |
| Ethics oversight | n/a |

Note that full information on the approval of the study protocol must also be provided in the manuscript.

# Field-specific reporting

Please select the one below that is the best fit for your research. If you are not sure, read the appropriate sections before making your selection.

☐ Life sciences   ☐ Behavioural & social sciences   ☒ Ecological, evolutionary & environmental sciences

For a reference copy of the document with all sections, see nature.com/documents/nr-reporting-summary-flat.pdf

# Ecological, evolutionary & environmental sciences study design

All studies must disclose on these points even when the disclosure is negative.

| | |
|---|---|
| Study description | This study addresses monitoring of population genetic diversity and the detection of potential genetic effects of climate change. We collected comprehensive information on monitoring projects for population genetic diversity (PGD), for purposes of management and conservation, in EU-COST program full-participant countries. The monitoring projects were tallied by various functional and taxonomic species groups, and by country.   The number of PGD monitoring projects in each country was used as an indicator of genetic monitoring effort (GME) for PGD monitoring. We modeled GME as a function of fundamental economic and geographical descriptors to identify factors associated with differences in monitoring activity among countries. We also modeled and mapped core and marginal conditions of the species climate niches (climate niche marginality), under current and potential future climatic conditions, for four groups of species of current and/or potential future conservation interest: Amphibians, and selections of large birds, carnivorans, and  forest trees.  We produced maps of these distributions in COST countries, and additionally mapped two derived variables: (i) the increases in the number of species newly experiencing marginal niche conditions, (ii) numbers of study species experiencing any loss of area with suitable climate conditions, as indicators of environmental degradation associated with predicted climate change. These values were combined across species to map the changing distribution of species joint niche marginality in COST countries that is associated with ongoing climate change.  Country values of current and predicted future joint niche marginality were plotted against country GME to infer how countries vary in need for monitoring and effort (and potentially capacity) to monitor, identify, and manage the genetic impacts of ongoing climate change on populations of conservation interest. |
| Research sample | We collected comprehensive data on projects monitoring population genetic diversity for conservation and management purposes in COST-member countries, constituting a census of such projects across all COST full-member countries, as of the start of data collection. These data were gathered using professional and stakeholder networks, and through a structured search of the Web of Science. We also acquired range and occurrence data on selected species in four target groups: European amphibians, and selected large birds, carnivorans, and forest trees.  These species were selected to comprise a set of species that represents current and potential future conservation and management interest across all COST full-member countries, regardless of EU membership. Range maps originated from the IUCN and occurrence data from the Global Biodiversity Information Facility.  We also acquired climate data for the current period and for a future period spanning the years 2041-2070 from the Chelsa database vers. 2.1.  These were chosen in order to estimate current and future climate niche marginality for the target species.  Finally, we used landcover/landuse data from the CORINE 2018 layer in order to filter the IUCN range maps for inappropriate habitat types, thereby refining our estimate of species current distributions. |
| Sampling strategy | We made every effort to census genetic monitoring projects, constructing a comprehensive data set.  Target species were selected to represent taxonomically divergent species of known or potential future conservation and management interest. |
| Data collection | Data on candidate projects for evaluation as Category II projects that monitor population genetic diversity over time, as defined by a peer-reviewed, published source, were collected through professional networks centered on population geneticists and practitioners involved in the COST Action 'Genomic Biodiversity Knowledge for Resilient Ecosystems (G-BiKE, https://www.cost.eu/actions/CA18134/)'.  This was done to discover monitoring efforts that had not necessarily appeared in the peer-reviewed literature. Additionally, a search of the Web of Science was conducted using broad search terms and a selection of relevant journals. |

| | |
|---|---|
| Timing and spatial scale | Data were collected between 10.2019 and 31.21.2021.  Data from the Web of Science were collected during 12.2021.  Additional information was collected during 1.1.2022-30.4.2022 to identify substantiating documentation dated prior to 1.1.2022. |
| Data exclusions | Candidate monitoring projects and those gathered through search of the Web of Science were evaluated for validity as Category II monitoring projects, following Schwarz et. al 2007. TREE 22:25-33.  Candidate projects not passing this validation and projects without samples from at least one COST full-member country were excluded. |
| Reproducibility | Validity of submitted, candidate projects for monitoring population genetic diversity was evaluated independently by at least two persons. When consensus was not reached, the first and/or last author served as a tie-breaking opinion. |
| Randomization | Each submitted candidate project was evaluated for validity (following Schwarz et al 2007) by randomly assigning the project to two evaluators who worked independently, then attempted to reach a consensus opinion. |
| Blinding | none |

Did the study involve field work? ☐ Yes ☒ No

# Reporting for specific materials, systems and methods

We require information from authors about some types of materials, experimental systems and methods used in many studies. Here, indicate whether each material, system or method listed is relevant to your study. If you are not sure if a list item applies to your research, read the appropriate section before selecting a response.

### Materials & experimental systems

| n/a | Involved in the study |
|---|---|
| ☒ ☐ | Antibodies |
| ☒ ☐ | Eukaryotic cell lines |
| ☒ ☐ | Palaeontology and archaeology |
| ☒ ☐ | Animals and other organisms |
| ☒ ☐ | Clinical data |
| ☒ ☐ | Dual use research of concern |
| ☒ ☐ | Plants |

### Methods

| n/a | Involved in the study |
|---|---|
| ☒ ☐ | ChIP-seq |
| ☒ ☐ | Flow cytometry |
| ☒ ☐ | MRI-based neuroimaging |

