## [Peer Review File · Nature Ecology & Evolution]

Peer Review Information

Journal: Nature Ecology & Evolution

Manuscript Title: Monitoring species genetic diversity in Europe varies greatly and overlooks potential climate change impacts

Corresponding author name(s): Peter B. Pearman

Editorial Notes:

Reviewer Comments & Decisions:

Decision Letter, initial version:

5th May 2023

Dear Professor Pearman,

Your manuscript entitled "Conserving genetic diversity during climate change: Niche marginality and discrepant monitoring capacity in Europe" has now been seen by 3 reviewers, whose comments are attached. The reviewers have raised a number of concerns which will need to be addressed before we can offer publication in Nature Ecology & Evolution. We will therefore need to see your responses to the criticisms raised and to some editorial concerns, along with a revised manuscript, before we can reach a final decision regarding publication.

As you can see from the reports, although Referees #1 and #2 were largely positive, Referee #3 had a number of criticisms that you will need to respond to -- including the appropriate use of the term 'capacity' in this context, and the use of niche marginality as an appropriate indicator of adaptive potential.

On a more personal note, we at Nature Ecology and Evolution would, like Referee #1, like to express our sincere condolences at the recent loss of your colleague Mike Bruford. I was taught conservation biology by Mike as an undergraduate, and always found his lectures fascinating, and I know he was deeply respected in the community.

We therefore invite you to revise your manuscript taking into account all reviewer and editor comments. Please highlight all changes in the manuscript text file [OPTIONAL: in Microsoft Word format].

2* If you have not done so already please begin to revise your manuscript so that it conforms to our Article format instructions at <http://www.nature.com/natecolevol/info/final-submission>. Refer also to any guidelines provided in this letter.

[REDACTED]

Nature Ecology & Evolution is committed to improving transparency in authorship. As part of our efforts in this direction, we are now requesting that all authors identified as 'corresponding author' on published papers create and link their Open Researcher and Contributor Identifier (ORCID) with their account on the Manuscript Tracking System (MTS), prior to acceptance. ORCID helps the scientific community achieve unambiguous attribution of all scholarly contributions. You can create and link your ORCID from the home page of the MTS by clicking on 'Modify my Springer Nature account'. For more information please visit www.springernature.com/orcid.

[REDACTED]

Reviewer expertise:

Reviewer #1: Conservation genetics, applied ecology, wildlife monitoring

Reviewer #2: Niche limits, genetics, distributions

Reviewer #3: Genetic diversity, conservation monitoring

Reviewers' comments:

2Reviewer #1 (Remarks to the Author):

I was deeply saddened by the news of Michael Bruford's passing, and would like to express my condolences and sympathy to the authors' team, who lost their colleague, one of the greatest evolutionary biologists of our time. I also apologize for the delay with submitting my review.

The manuscript by Pearman and colleagues represents a great contribution to evolutionary and conservation biology. The authors address the critical issue of the need to monitor population genetic diversity, especially in places where shifts of species ranges are expected based on climate predictions, in order to be able to predict future changes of genetic diversity and manage impacts of climate change on populations. The authors use innovative approach to integrate the data on margins of species distributions across 185 species and existing monitoring programs to identify areas and taxonomic groups in Europe that urgently require development of genetic monitoring capacity. The amount of work that went into identifying and filtering monitoring projects across Europe is impressive. It was also interesting to see analysis of factors which can influence countries' capacity to conduct genetic monitoring. The focus on niche margins, which are most vulnerable to climate change and important for adaptation, also helped authors to focus on the most problematic areas. Given that genetic monitoring is currently geographically and taxonomically limited and not well integrated into global biodiversity monitoring efforts, there is a need in focusing resources on developing it in countries with high predicted climate-driven decline in habitat suitability. The authors' findings will aid governments of European countries in allocating resources towards areas and groups that require genetic monitoring, and also help evolutionary and conservation biologists target their future studies.

I enjoyed reading this paper- the explanations are clear, methods are sound and mostly clearly explained, sample sizes are respectable, and quality of graphics is great. I did not spot any serious flaws. All my comments and questions are relatively minor and mainly concern clarity.

I had some questions. I was wondering whether and how the authors accounted for the species richness? Are there more monitoring studies in countries with highest species richness and which countries are lagging behind? I also wasn't clear on whether Turkey doesn't conduct genetic monitoring or just does not conduct/report the monitoring studies in the specific way required for this study? If the latter is true, is it fair to include it in the study for comparisons with other COST countries? But I understand that including it allowed the authors to highlight it as an area in need of increased monitoring capacity.

I also found it challenging to think about the biological meaning of the change in niche marginality. The location can stop being marginal for a species if this species is no longer present (extinct) in the area, or if the climate change make this location less marginal and more core, something you allude to in L487. As these are outcomes of different processes, the change per se does not have an intuitive interpretation. Is it worth differentiating between the two outcomes?

Other minor comments:

Abstract: Perhaps add a sentence clearly formulating why this research was important to conduct. I found a clear statement on L332
L203-205 Please rephrase, the sentence is confusing. 'Climate niche marginality' needs to be defined
L206 replace 'and' with full stop.

3L226 accounting while doing what?

L237 endangered by what? through which process? Please clarify

L242 'to detect PGD'- perhaps to detect changes in PGD?

L327-330 long sentence, please split.

L333-342 The authors may consider moving some essential explanations from methods to this section (or at least summarising them), as it was difficult to grasp on the first read. What 'joint geographic distribution of niche marginality' represents and how it was summarized was not clear.

L286 Why striking north-south pattern was expected? Was it based on countries' economies? Or on number of species?

L290: 38-31=7, but Fig.1b indicates 8 countries with no GMC.

L351-353 How does this relate to the total number of (selected) species per country?

L369 May be I am missing the point, but how this statement differs from that on L351-353?

L383 Why do you describe an observation as a prediction?

L393 Can you call here for non-COST countries adopting the use of Category II monitoring protocols?

L397 It would be good to discuss or mention species richness here. Is this one of the possible characteristics?

L424 Is something missing after 'thus'? How one is to account for climate change in the design? Do you mean choosing appropriate markers?

L427-440 I am wondering if it is realistic and useful to expect countries to monitor populations at 'functional loci'. There is enough evidence that evolution of novel adaptation can involve multiple loci, and regulatory regions, rather than protein sequences, and that genome-wide genetic diversity could be a reasonably proxy of adaptive potential (Harrisson et al. 2014 Evol App). Recommending monitoring diversity at 'functional' and 'neutral' loci means that before being able to monitor PGD researchers have to establish 'functional loci' relevant to each species. Of course this requires time and could be considered too expensive by some countries with low GDP and high species richness. It would be good to know how authors suggest to approach this (same question for L469). With cost of sequencing going down, why not just recommend monitoring genome-wide diversity?

L455 Would it be useful to call for creation of international initiatives for genetic monitoring, financed through EU or COST?

L468 and baseline genetic diversity?

L479 Shouldn't these be species with range-wide baseline genetic data available, and not just data for marginal niche locations? Genetic responses may differ depending on species range, mobility, baseline genetic diversity etc.

L512 May be I misunderstood, but is it possible to distinguish retreating niche margins from advancing niche margins from your maps?

L686 I did not completely understand how the projection into climate space was done for each species and what are species scores. May be you could add a figure to illustrate what you have done on one example species?

L782 remove 'of'

Table 1 is not cited in the manuscript. If such a citation is not needed, perhaps it could be moved to Online Methods? Some variables require explanation (e.g. 'Frequency (annual?)'- does it mean that monitoring done twice 10 years apart will get 'FALSE'? 'European Union Directive and Annex'- what is it?).

Fig. 1 Adding country codes to Fig 1a would help the reader unfamiliar with postal codes to better understand geographic context of Fig. 3 and Fig. 6

Fig.5, Extended Data Figure 4. Comparing the two maps is difficult- could the third panel of maps show the difference between current and future numbers?

Online Methods Figure 1- Is this process for selecting Category II monitoring projects? If so, please

mention 'Category II monitoring' in the legend. Why question 6 is needed if both outcomes yield to the same outcome (project rejected)? Would it be worth including this figure in the main ms?

Reviewer #2 (Remarks to the Author):

I read this manuscript with interest, as it tackles the fundamental problem of whether we are collecting sufficient genetic data across species ranges to make appropriate inferences about this cryptic layer of biodiversity. My review will be short, but it is not for lack of engagement with the manuscript. I found the manuscript to be exceedingly clear, the analyses are robust, and findings are straightforward.

My only major comment is that I did not follow the logic of requiring studies to have two distinct time points for repeated sampling. Why would a single snapshot of genetic variation across a species range not suffice to make inferences about the spatial distribution of genetic variation and how that relates to climatic niche marginality? Other than this one concern, however, I found no fault with other aspects of the design and analysis choices.

Reviewer #3 (Remarks to the Author):

Pearman and colleagues aim to investigate the landscape of genetic diversity monitoring programs in Europe and predict the countries and areas that require increased monitoring efforts based on niche marginality analysis. Monitoring genetic diversity is crucial and hence an important topic, but from my perspective, the manuscript needs several updates and refinements to be considered for publication in Nature Ecology and Evolution.

The term "capacity" used in the manuscript is highly misleading. The authors report only the number of ongoing monitoring programs per country, hence 'efforts', rather than the actual capacity of each country for carrying out monitoring projects. This term needs to be adjusted to reflect what is shown in the manuscript.

The recent CBD post-2020 global biodiversity framework discussions (Hoban et al. 2020; Laikre et al. 2020; Thurfjell et al. 2022; Hoban et al. 2023), as well as the new CBD COP15 Goal A and Target 4 decisions, have had a significant impact on the European genetic diversity monitoring landscape. There are now many more ongoing monitoring programs than during the period recorded in the manuscript (2019 to 2021). Additionally, most of the information is based on the European Commission's Cooperation in Science and Technology (COST) program, which may cause some bias. Moreover, the manuscript refers to 38 member countries, but there are currently 41 members. The authors' selection of monitoring programs and species sounds sophisticated but also shows bias towards the COST network.

Furthermore, the use of niche marginality as a good indicator of adaptive potential remains highly questionable. Such marginal populations usually have lower genetic diversity and are adapted to specific environments (Dauphin et al. 2020), making them highly vulnerable to any environmental change. Marginality was calculated based on 19 bioclimatic variables. If a site belongs to the 25% most climatically marginal sites, it was defined as marginal. This means that there are marginal habitats for both low and high temperatures. However, for instance, low temperatures will not carry any adaptive diversity for future climate change. The models and assumptions used in the analysis should be more specific and refined.

5Moreover, the researchers utilized the most extreme climate model (SSP5-8.5) to forecast the forthcoming alteration in marginality without providing any rationale and without conducting a comparison with other models. Furthermore, it was not specified which of the five models available for the time span 2051-70 at

https://envicloud.wsl.ch/#/?prefix=chelsa/chelsa_V2/GLOBAL/climatologies/ was selected, nor was the rationale for the choice provided. It should be noted that the different models exhibit varying levels of climate sensitivity, and therefore the selection of a particular model is significant. This approach may raise questions about the robustness and generalizability of their findings.

The manuscript completely ignores insects, which are currently suffering a significant die-off. Hence, the selection of species for the marginality analysis appears arbitrary, and many relevant species from different ecosystems are missing.

In conclusion, several critical considerations need to be addressed and refined in the manuscript to meet the standards of Nature Ecology and Evolution.

More detailed comments to specific regions of the manuscript are given below.

--Title--

As mentioned above the term 'capacity' is very misleading in the context reported in the manuscript

--Abstract--

>>Line 195ff and key words: The term 'genetic monitoring' encompasses monitoring programs that utilize environmental DNA (eDNA) approaches. However, these programs are primarily focused on inter-species diversity, rather than monitoring intra-species genetic diversity. Given the rapid development of various fields, it is important to ensure the specificity of definitions.

--Introduction--

>> Line 222ff: I disagree with the general statement that populations in extreme climatic conditions, such as those near their climatic niche margins, are particularly relevant to species potential for adaptation to changing climate. Such marginal populations usually have lower genetic diversity and are adapted to specific environments (Dauphin et al. 2020), making them highly vulnerable to any environmental change.

I respectfully disagree with the broad assertion that populations in extreme climatic conditions, such as those near their climatic niche margins, are particularly relevant to species potential for adaptation to changing climate. Typically, populations situated at the edge of their ecological ranges tend to exhibit lower genetic diversity and are adapted to specific environmental conditions (Dauphin et al. 2020), because they lost a lot of genetic diversity during the colonization process (Wegmann et al. 2006; Takahashi et al. 2016) and adaptation (Ellegren & Galtier 2016). Consequently, these populations can be particularly susceptible to the impacts of environmental changes.

>> Line 233ff: The authors have stated that populations inhabiting areas near the limits of warm and dry habitats possess significant adaptive genetic diversity. However, their analysis of niche marginality does not specifically target this particular marginal habitat. Instead, it encompasses a broader range of marginal niches on a global scale. Hence, their reasoning and performed analysis are not in line.

>>Line 247: The utilization of numerous abbreviations within the manuscript impairs its readability, particularly when the abbreviations PDG and GDP are utilized interchangeably, solely differing in the order of their letters. Furthermore, the abbreviation GMC may require alteration since it fails to accurately represent the shown data.

--Results--

>>Line 278: At present, there exist 41 countries affiliated with COST, as opposed to the here stated 38, indicating that the manuscript is outdated.

>> Line 296: The COST specification indicating that the results are specific to COST countries may give the impression to the reader that the manuscript is intended solely for COST countries and may have limited relevance to a wider audience.

>>Line 351: Assuming that species' climate niche remains stable over time is a strong assumption, and furthermore, it has been demonstrated that species distribution models can vary even within species (Razgour et al. 2019; Chardon et al. 2020)

>>Line 371: Overall, I am skeptical about the meaningfulness of summarizing niche marginality across different species, as most species have highly specific niches, as is well-known in amphibians (Rodriguez-Rodriguez et al. 2020).

--Discussion--

>>Line 404: I disagree with the statement that only a small fraction of studies have been missed. In recent years, the research field has rapidly evolved, and many more projects have been initiated since the recording phase (2019-2021). Additionally, the strong focus on the COST countries/project introduced another potential bias.

>>Line 511: The author claims that 'The present study suggests that populations towards the warm/dry, retreating niche margins are geographically clustered in Europe.' However, given the presented results, I cannot follow this argument as marginality was not calculated for dry/warm niches.

--Materials and Methods--

>>Line 703: It is unclear why the most extreme climate model (SSP5-8.5) was used to forecast forthcoming alterations in marginality, and additionally it was not specified which of the five models available for the time span 2051-70 at https://envicloud.wsl.ch/#/?prefix=chelsea/chelsea_V2/GLOBAL/climatologies/ was selected. It should be noted that the different models exhibit varying levels of climate sensitivity, and therefore the selection of a particular model is significant. This approach may raise questions about the robustness and generalizability of the findings.

-- Figures--

>>Figure 6: As the shown "capacity" does not reflect the actual potential, it is unclear what this figure can tell us.

>>Extended Data Figure 5: The legend at the top of the figure and the text legend do not match, making it difficult to understand the content of the figure.

References mentioned in the reply letter

Chardon NI, Pironon S, Peterson ML, Doak DF (2020) Incorporating intraspecific variation into species distribution models improves distribution predictions, but cannot predict species traits for a wide-

spread plant species. *Ecography* 43, 60-74.

Dauphin B, Wüest RO, Brodbeck S, Zoller S, Fischer MC, Holderegger R, Gugerli F, Rellstab C (2020) Disentangling the effects of geographic peripherality and habitat suitability on neutral and adaptive genetic variation in Swiss stone pine. *Molecular Ecology* 29, 1972-1989.

Ellegren H, Galtier N (2016) Determinants of genetic diversity. *Nature Reviews Genetics* 17, 422-433.

Hoban S, Bruford M, D'Urban Jackson J, Lopes-Fernandes M, Heuertz M, Hohenlohe PA, Paz-Vinas I, Sjögren-Gulve P, Segelbacher G, Vernesi C, Aitken S, Bertola LD, Bloomer P, Breed M, Rodríguez-Correa H, Funk WC, Grueber CE, Hunter ME, Jaffe R, Liggins L, Mergeay J, Moharrek F, O'Brien D, Ogden R, Palma-Silva C, Pierson J, Ramakrishnan U, Simo-Droissart M, Tani N, Waits L, Laikre L (2020) Genetic diversity targets and indicators in the CBD post-2020 Global Biodiversity Framework must be improved. *Biological Conservation* 248.

Hoban S, Bruford MW, da Silva JM, Funk WC, Frankham R, Gill MJ, Grueber CE, Heuertz M, Hunter ME, Kershaw F, Lacy RC, Lees C, Lopes-Fernandes M, MacDonald AJ, Mastretta-Yanes A, McGowan PJK, Meek MH, Mergeay J, Millette KL, Mittan-Moreau CS, Navarro LM, O'Brien D, Ogden R, Segelbacher G, Paz-Vinas I, Vernesi C, Laikre L (2023) Genetic diversity goals and targets have improved, but remain insufficient for clear implementation of the post-2020 global biodiversity framework. *Conservation Genetics*.

Laikre L, Hoban S, Bruford MW, Segelbacher G, Allendorf FW, Gajardo G, Rodríguez AG, Hedrick PW, Heuertz M, Hohenlohe PA, Jaffé R, Johannesson K, Liggins L, MacDonald AJ, Orozco-Wengel P, Reusch TBH, Rodríguez-Correa H, Russo I-RM, Ryman N, Vernesi C (2020) Post-2020 goals overlook genetic diversity. *Science* 367, 1083.

Razgour O, Forester B, Taggart JB, Bekaert M, Juste J, Ibáñez C, Puechmaile SJ, Novella-Fernandez R, Alberdi A, Manel S (2019) Considering adaptive genetic variation in climate change vulnerability assessment reduces species range loss projections. *Proceedings of the National Academy of Sciences* 116, 10418-10423.

Rodríguez-Rodríguez EJ, Beltran JF, Tejedo M, Nicieza AG, Llusia D, Marquez R, Aragon P (2020) Niche models at inter- and intraspecific levels reveal hierarchical niche differentiation in midwife toads. *Sci Rep* 10, 10942.

Takahashi Y, Suyama Y, Matsuki Y, Funayama R, Nakayama K, Kawata M (2016) Lack of genetic variation prevents adaptation at the geographic range margin in a damselfly. *Molecular Ecology* 25, 4450-4460.

Thurfjell H, Laikre L, Ekblom R, Hoban S, Sjögren-Gulve P (2022) Practical application of indicators for genetic diversity in CBD post-2020 global biodiversity framework implementation. *Ecological Indicators* 142.

Wegmann D, Currat M, Excoffier L (2006) Molecular diversity after a range expansion in heterogeneous environments. *Genetics* 174, 2009-2020.

*****END*****

Author Rebuttal to Initial comments

Referee 1.

1. I had some questions. I was wondering whether and how the authors accounted for the species richness? Are there more monitoring studies in countries with highest species richness and which countries are lagging behind?

Response: Thank you for this question. Simply put, we did not model genetic monitoring as a function of national measures of species richness. However, it is widely known that there is a north-south richness gradient for many higher taxa in Europe. For instance, European hotspots of biodiversity are situated in the South of the continent (Myers et al 2000, Nature 403:853-858). Based on this, if it were reasonable to expect more monitoring programs in countries with greater species richness, one would expect more projects in southern countries. This is generally not the case, since Turkey has no qualifying programs and Sweden, with lower species richness in many taxa, has one of the highest counts of monitoring programs. Further, the Balkan peninsula is also recognized as having high species richness (Neubert et al 2019; Freyhof and Books 2011), and countries in this region have very few projects to monitor population genetic diversity. In response to this comment, we have created a new figure of the species richness of study species in COST countries (Extended Data Fig.4). We clarify on lines 392-394 that we observe no North-South gradient in country monitoring efforts.

2. I also wasn't clear on whether Turkey doesn't conduct genetic monitoring or just does not conduct/report the monitoring studies in the specific way required for this study? If the latter is true, is it fair to include it in the study for comparisons with other COST countries?

Response: Thank you for this question and we are happy to clarify this issue. The 'specific way' required by this study for counting genetic monitoring programs was some form of written documentation of Category II monitoring (Schwartz et al. 2007). We found no documentation of any programs in Turkey, following our criteria. Our criteria were applied to all COST countries

(including Turkey) in a consistent, uniform manner, as specified in the Methods section. Please see lines 732-761. The available information does not suggest any bias in our criteria, or which could have excluded monitoring efforts in any country.

3. But I understand that including it allowed the authors to highlight it as an area in need of increased monitoring capacity.

Response: We are happy to clarify this further. We included Turkey because it is a full member in the COST program, therefore meeting the criteria for inclusion, and for no other reason.

4. I also found it challenging to think about the biological meaning of the change in niche marginality. The location can stop being marginal for a species if this species is no longer present (extinct) in the area, or if the climate change makes this location less marginal and more core, something you allude to in L487. As these are outcomes of different processes, the change per se does not have an intuitive interpretation. Is it worth differentiating between the two outcomes?

Response: We thank the referee for pointing this out. We have designed new maps at the species level to distinguish between areas with leading marginal and trailing marginal climate, and we now provide two maps for each species in greatly expanded supplementary materials. These maps are presented for multiple emission scenarios and global circulation models. For each species we present maps of the current and future geographic distribution of leading and trailing marginal areas within the current species range. Using vectors, we illustrate several kinds of changes that are predicted under each scenario/GCM combination (please see the new Figure 4 and Appendices S3-S6, Supplementary Materials). Determination of these areas and the categories of environmental (niche) change are now described in the online methods section in detail (see lines 900-903). An additional figure for the main text now provides a specific example, with maps in the same format as those in the supplementary materials (please see Figure 4). We believe that these changes have clarified niche marginality, its calculation, and the potential effects of predicted climate change on its geographic distribution at the species level. In the subsequent analyses we present, we focus exclusively on trailing-edge niche marginality.

Other minor comments:

5. Abstract: Perhaps add a sentence clearly formulating why this research was important to conduct. I found a clear statement on L332

Response: A statement of the importance of the research is now integrated into the Abstract as follows: "We report the first accounting of genetic monitoring efforts among countries in Europe ('genetic monitoring effort', GME), an evaluation of which can help guide future capacity building and collaboration efforts to areas most in need." See lines 283-286.

6. L203-205 Please rephrase, the sentence is confusing. 'Climate niche marginality' needs to be defined

Response: We thank the referee for pointing out the awkwardness here. We think that providing a definition of climate niche marginality in the Abstract would be cumbersome. We respond to the referee's concern by avoiding definition of new jargon in the abstract and instead write "Our analysis suggests that country area extent, financial resources, and conservation policy influence GME, high values of which only partially match species joint patterns of limits to suitable climatic conditions." This can be found on lines 290-291. We define trailing-edge niche marginality later in the text (lines 434-435) and describe the method of its calculation (lines 900-903).

10

ess
is

7. L206 replace ‘and’ with full stop.

Response: Changed as requested. See new line number 292.

8. L226 accounting while doing what?

Response: We have edited the sentence to read: “This suggests the need for improved quantification of the relationships between species limits along environmental gradients and associated PGD^{17,18} “ Please see lines 315-317.

9. L237 endangered by what? through which process? Please clarify

Response: What we meant was that genetic diversity in marginal populations may be lost. We have modified the sentence to read: “Nonetheless, genetic diversity and adaptive variants held in marginal populations may be lost (i) when gene flow to environmentally central areas is impeded, (ii) due to genetic drift strongly affecting populations with small effective populations sizes, or (iii) if the populations go extinct as climate extremes eventually exceed species tolerances²⁸.” Please see lines 326-330.

10. L242 ‘to detect PGD’- perhaps to detect changes in PGD?

Response: Yes, since we are writing about genetic monitoring, inserting ‘changing’ is appropriate. Please see the new line 335.

11. L327-330 long sentence, please split.

Response: Done. The sentences now read “Countries with a relatively large GME should be well prepared to evaluate climate impacts on genetic diversity because they have much relevant infrastructure (i.e., genetic laboratories). Additionally, some aspects of adapting monitoring programs to detect effects of climate change are technically simple, such as expanding sampling to cover climate gradients.” Please see lines 580 – 584.

12. L333-342 The authors may consider moving some essential explanations from methods to this section (or at least summarizing them), as it was difficult to grasp on the first read. What ‘joint geographic distribution of niche marginality’ represents and how it was summarized was not clear.

Response: We agree that the presentation was a bit over-concise. The presentation has been expanded and re-organized to make it easier to understand, while we also try not to inflate the word count unduly. We believe the resulting text is clearer. We leave the detailed explanation for the online Methods section. Please see lines 355-361 in the main text.

13. L286 Why striking north-south pattern was expected? Was it based on countries’ economies? Or on number of species?

Response: We meant ‘obvious’ or ‘notable’ and have edited this sentence to remove ‘striking’. Further, we initially had no strong expectations of the relationship between GME and latitude. As noted by the referee, one might expect higher GME in the north of Europe, where GDP is higher, or in the south of Europe because of its higher habitat diversity, species endemism and biodiversity hotspots (Myers et al. 2000 Nature, Cervellini et al. 2021 Ecology and Evolution). We previously introduced the issues of GDP and GME. We have added one sentence to the Introduction to express

11

ess
is

the expectation that because of greater species endemism and biodiversity hotspots, one might expect greater GME in southern Europe. See lines 352-353 and 392-394.

14. L290: 38-31=7, but Fig.1b indicates 8 countries with no GMC.

Response: Thank you. That is our mistake. The correct number of countries with positive GME values is 30 and eight countries have had no PGD monitoring. Please see line 396.

15. L351-353 How does this relate to the total number of (selected) species per country?

Response: We believe the reviewer is asking about the total number of species per country in order to determine the proportion of species. We answer by emphasizing why this test was performed. The analysis was performed to highlight the overall monitoring effort that might be needed per country, for marginal populations (Fig 5), and to examine an estimate of this need in the future. The figure serves as a national summary of the purely geographic distribution exemplified in Fig. 6, and provides a basis for geographic comparisons among countries, which are made explicit graphically in Fig. 7 and Appendix S9. Our results in Figure 5 show that some countries have more species for which monitoring programs are needed, essentially, that they should be monitoring species because of the number threatened by climate change. As noted in the discussion, this need does not align with current and recent genetic monitoring efforts. As can be seen by examining Figs. 1, 5 and 6, effort and need appear aligned (high monitoring = high need, low monitoring = low need) for some countries (France, Spain, Italy, some but not all of Balkans) but not others (most of northern Europe, Turkey, parts of Balkans). The conclusion is that although there is high capacity for genetic monitoring in northern Europe, for example, that does not match to a relatively high need for genetic monitoring in the context of climate change. And there is a high need in Turkey, but low documented effort (and perhaps low capacity). The important point for the referee might be that the 'need' for monitoring is a function of number of species rather than proportion. For any stakeholder (government, scientist, conservationist), the cost of genetic monitoring increases with the number of species, not the proportion. Management agencies will most be concerned with the number of species needing to be genetically monitored as that is needed for budgeting, planning, reporting etc. The proportion is not easily translated to 'need' as the actual number of species.

16. L369 Maybe I am missing the point, but how does this statement differ from that on L351-353? L383 Why do you describe an observation as a prediction?

Response: The sentence was superfluous and was deleted.

17. L393 Can you call here for non-COST countries adopting the use of Category II monitoring protocols?

Response: We add the following sentence to the Discussion: "Category II monitoring programs that span climatic gradients occupied by species should be established in additional countries not involved in the COST program, wherever climate analysis suggests increasing niche marginality of populations of conservation interest, or a risk of genetic erosion is suspected." Please see lines 660-663.

18. L397 It would be good to discuss or mention species richness here. Is this one of the possible characteristics?

Response: We edit and expand the text of the first paragraph of the Discussion, to include an important aspect along these lines, as follows: "...and many factors conceivably influence the

12

ess
is

establishment of monitoring programs, regardless of country size or per capita GDP, such as the number of wild species of traditional or cultural importance, or species richness.” Please see lines 532 – 535.

19. L424 Is something missing after ‘thus’? How one is to account for climate change in the design? Do you mean choosing appropriate markers?

Response: No, we meant by establishing monitoring along climate gradients, which is addressed elsewhere in the manuscript, including in the Abstract (see lines 292-294). The sentence was out of place in this paragraph and has been removed.

20. L427-440 I am wondering if it is realistic and useful to expect countries to monitor populations at ‘functional loci’. There is enough evidence that evolution of novel adaptation can involve multiple loci, and regulatory regions, rather than protein sequences, and that genome-wide genetic diversity could be a reasonable proxy of adaptive potential (Harrison et al. 2014 Evol App). Recommending monitoring diversity at ‘functional’ and ‘neutral’ loci means that before being able to monitor PGD researchers have to establish ‘functional loci’ relevant to each species. Of course, this requires time and could be considered too expensive by some countries with low GDP and high species richness. It would be good to know how authors suggest to approach this (same question for L469). With cost of sequencing going down, why not just recommend monitoring genome-wide diversity?

Response: We agree with the referee. While studies to identify candidate loci for adaptation are now fairly common, we agree that genome-wide data, when feasible, would provide comprehensive information about the time course and structure of both adaptively important and neutral variation. We now call for estimation of genome-wide diversity, with an appropriate citation. Please see lines 588 – 592.

21. L455 Would it be useful to call for creation of international initiatives for genetic monitoring, financed through EU or COST?

Response: It is unlikely that COST would finance genetic monitoring directly, since it focuses on developing scientific networks. We have added text to the Abstract: “This need could be met in part by expanding the European Union’s Birds and Habitats Directives to fully address conservation and monitoring of genetic diversity” (lines 294-296), and to the last paragraph of the Discussion: “Our results may guide future EU investment in genetic monitoring programs and in conservation genetic research projects. Positive developments in the support of PGD monitoring that have arisen from COP 15 can be leveraged by adopting language in the European Union’s Bird and Habitats Directives to support genetic monitoring. Similar actions should be taken by governments outside of Europe. Future projects may also focus networking and training efforts more strongly in certain regions, such as the Balkan countries and Turkey”. Please see lines 681-686.

22. L468 and baseline genetic diversity?

Response: We agree that this is a good suggestion. Added. Please see lines 627-630.

23. L479 Shouldn’t these be species with range-wide baseline genetic data available, and not just data for marginal niche locations? Genetic responses may differ depending on species range, mobility, baseline genetic diversity etc.

13

ess
is

Response: We did not mean to imply that species should only be monitored in areas near niche margins, and we have stated elsewhere that monitoring should include samples along the entire climate gradient where the species is found. We have tried to clarify this at the point in the text mentioned by the referee to avoid confusion. Please see lines 639-642.

24. L512 Maybe I misunderstood, but is it possible to distinguish retreating niche margins from advancing niche margins from your maps?

Response: Yes, it is possible and has now been done for new species maps in Appendices S3-6 in Supplementary Materials and in the new Figure 4. Please see lines 900-903.

25. L686 I did not completely understand how the projection into climate space was done for each species and what are species scores. Maybe you could add a figure to illustrate what you have done on one example species?

Response: We thank the referee for this suggestion. We have added a new figure to illustrate these relationships. Please see Figure 4 and references to it, lines 879-884 and 900-903.

26. L782 remove 'of'

Response: This has been done and the Author Contribution section has been updated to improve the attribution of various contributions. Done as requested. Please see lines 991-999.

27. Table 1 is not cited in the manuscript. If such a citation is not needed, perhaps it could be moved to Online Methods? Some variables require explanation (e.g. 'Frequency (annual?)'- does it mean that monitoring done twice 10 years apart will get 'FALSE'? 'European Union Directive and Annex'- what is it?).

Response: Table 1 is now cited in the first paragraph of the Results section and in the second paragraph of the online methods section. The text has been expanded to clarify the issues identified by the referee. Please see lines 371-372 and 725. This table was for data collection and presents no evaluative criteria. The raw monitoring data are presented in Appendix S11. Please see lines 1002-1004. We expand the text slightly to refer to both the Habitats and Bird Directives, which are the primary conservation instruments of the European Union. Please see revised Table 1.

28. Fig. 1 Adding country codes to Fig 1a would help the reader unfamiliar with postal codes to better understand geographic context of Fig. 3 and Fig. 6

Response: We considered several systems of country codes. Adding codes to the figure appears infeasible because the size of many countries is so small that the letters will not be legible. For this reason, we now integrate the codes and corresponding country names into Figs. 3 and 7.

29. Fig.5, Extended Data Figure 4. Comparing the two maps is difficult- could the third panel of maps show the difference between current and future numbers?

Response: We thank the referee for this suggestion and have considered it carefully. We agree with the referee that developing understanding of how niche marginality changes over the period is key to the development of the manuscript. We have addressed this comprehensively in re-working many of the figures and in several new figures. In the new Figs 5 and 6 we provide separate maps of the number of species that newly experience trailing-edge marginal niche conditions (Fig. 5e,f; Extended Data Figure 5c,f,i,l; Appendix S7), and loss of suitable habitat (Fig5g,h; Extended Data Figure 6c,f,i,l). Together, these new figures present two aspects of change in trailing edge

14

ess
is

marginality for the four groups of species and all species taken together. We also present the new Extended Data Fig. 7 that compares the difference between future and current marginality for COST countries. The effect of differing SSP-GCM combinations can be observed by examining the 'chng' maps in Appendix S7, Supplementary Materials). We have also plotted these values against Effort and present them in Appendix S9. Also see the README files associated with these appendices.

30. Online Methods Figure 1- Is this process for selecting Category II monitoring projects? If so, please mention 'Category II monitoring' in the legend.

Response: We have added reference to Category II monitoring in the figure legend.

31. Why question 6 is needed if both outcomes yield to the same outcome (project rejected)? Would it be worth including this figure in the main ms?

Response: While the information collected based on question six was not used as a classifier, it provides relevant background on the variation among the rejected projects. We therefore decided to keep question six in the flow chart. Since we have added a new figure to the main ms, we prefer not to add this one.

Referee 2.

I read this manuscript with interest, as it tackles the fundamental problem of whether we are collecting sufficient genetic data across species ranges to make appropriate inferences about this cryptic layer of biodiversity. My review will be short, but it is not for lack of engagement with the manuscript. I found the manuscript to be exceedingly clear, the analyses are robust, and findings are straightforward.

Response: Thank you for this supportive statement.

My only major comment is that I did not follow the logic of requiring studies to have two distinct time points for repeated sampling. Why would a single snapshot of genetic variation across a species range not suffice to make inferences about the spatial distribution of genetic variation and how that relates to climatic niche marginality?

Response: We agree that snapshots of genetic variation across a species range are helpful to set genetic baselines and can allow making some inferences about spatial genetic patterns and underlying processes. However, investigating the genetic consequences of rapidly-changing factors (e.g. climate-driven effects) with only present-day genetic data is tricky, as the lack of temporal perspective limits our capacity to disentangle the effects of multiple concomitant processes (Jensen & Leigh 2022 Ecol Evol 12:e9340; Clark et al. 2023, Mol Ecol Res 10.1111/1755-0998.13789 and references therein). Furthermore, a snapshot of the genetic variation levels observed in one or more populations is not sufficient as assessing and reporting on change in genetic diversity, which requires standardized genetic monitoring across multi-year time frames (Hoban et al, 2014. Evol Appl 7:984; Mathieu-Bégné et al., 2019. Oikos 128:196). The issue is that a snapshot is not monitoring because it does not establish a trajectory in genetic diversity over time. This is the basic property of monitoring programs for genetic diversity, as defined by Schwarz et al 2007. We use this definition in the development of this study. Please see lines 372-378.

15

ess
is

Referee 3.

Pearman and colleagues aim to investigate the landscape of genetic diversity monitoring programs in Europe and predict the countries and areas that require increased monitoring efforts based on niche marginality analysis. Monitoring genetic diversity is crucial and hence an important topic, but from my perspective, the manuscript needs several updates and refinements to be considered for publication in Nature Ecology and Evolution.

1. The term "capacity" used in the manuscript is highly misleading. The authors report only the number of ongoing monitoring programs per country, hence 'efforts', rather than the actual capacity of each country for carrying out monitoring projects. This term needs to be adjusted to reflect what is shown in the manuscript.

Response: We have substituted 'effort' in place of 'capacity' in the context of this usage, including in the title. We also now use GME instead of GMC, for Genetic Monitoring Effort.

2. The recent CBD post-2020 global biodiversity framework discussions (Hoban et al. 2020; Laikre et al. 2020; Thurfjell et al. 2022; Hoban et al. 2023), as well as the new CBD COP15 Goal A and Target 4 decisions, have had a significant impact on the European genetic diversity monitoring landscape. There are now many more ongoing monitoring programs than during the period recorded in the manuscript (2019 to 2021).

Response: Thank you for this comment. Due to the vetting process, the period for analysis, writing, review and further formatting, a study of an on-going process (here, genetic monitoring) will always miss the last few studies because it is not possible to include additional data points up to the moment of publication. For example, the influential genetic meta-analysis by Pinsky and Palumbi (2014 Mol Ecol 23:29-39) collected data till May 2011 but was first published online in Sept 2013. Likely a few more projects have been initiated, but it is unclear how many, which is something that can be addressed in future research that builds on our baseline. We know that COP 15 Goal A, Target 4 and the headline indicator A.4 of the Kunming-Montreal GBF monitoring framework were formally accepted in December 2022. We agree with the referee and hope that the new CBD GBF and recent discussions in the literature will have an impact on the European genetic diversity landscape, but it is unlikely that these developments have caused great numbers of new genetic monitoring projects, since few proposal-review-award cycles will have been completed since, if any. New genetic monitoring programs that might have been initiated since 31.12.2021 are most likely in countries that already have a strong history of monitoring, since they would be the ones with relevant experience and infrastructure. It is unlikely that many new programs have been initiated in countries lacking monitoring thus far. To support our contention that the available data are still current and relevant, one of us (PBP) repeated the Web of Science search on 16.07.2023, using the same search terms as were used for the manuscript, but with spanning dates of 1.1.2022-30.6.2023. This produced 30 hits, none of which qualified as Cat. II monitoring under the criteria we describe in the manuscript. This suggests that new PGD monitoring programs in the intervening months would not alter the trends we illuminate in the manuscript.

3. Additionally, most of the information is based on the European Commission's Cooperation in Science and Technology (COST) program, which may cause some bias. Moreover, the manuscript refers to 38 member countries, but there are currently 41 members. The authors' selection of monitoring programs and species sounds sophisticated but also shows bias towards the COST network.

16

ess
is

Response: Thank you for this comment and we appreciate the opportunity to respond to it. Ukraine, Georgia and Armenia officially entered into COST on 31.03.2022 and 10.11.2022 (Armenia), respectively, which were both after the end of the data collection period for this study. There were 38 COST full member countries during our data collection. We stress that although our study area covered most of Europe by focusing on COST full member countries at the time, the information included on genetic monitoring programs and species niche marginality are not directly based on genetic research resulting from COST program funding. The data contributors, who came from within and outside of the G-BiKE COST Action, were not instructed to only submit their own work, but rather to present candidate programs, projects or any other monitoring activity that they were aware of. There was no filtering of submissions at the point of contribution. The definition of monitoring that we adopted, following Schwartz et al. 2007, has nothing to do with the COST program. Restricting the study area to COST full member countries would not inherently cause bias, since COST includes almost all European countries, constituting a study area of nearly continental extent. A global study might have reached different conclusions than the ones we reach here, but the current scope of our study in terms of the number of countries is without precedent.

Nonetheless, we have modified the Discussion to include the statement that our study area is limited to COST full member countries up to 31.12.2021 and that our conclusions apply to this area. The implications, however, extend more broadly, to any country where genetic variation may be threatened by impacts of climate change. Please see lines 557-563, 660-663, and 683-686.

4. Furthermore, the use of niche marginality as a good indicator of adaptive potential remains highly questionable. Such marginal populations usually have lower genetic diversity and are adapted to specific environments (Dauphin et al. 2020), making them highly vulnerable to any environmental change.

Response: We thank the referee for the opportunity to clarify this issue. We agree with him/her that this is still an area of ongoing research (please see lines 576-578). However, neither we nor Dauphin et al. 2020 go so far as to claim that marginality indicates adaptive potential. The issue is focused on variation at loci likely involved in adaptation to environmentally marginal conditions. This is a subtle but important difference. Whether adaptation actually occurs as climate changes is a complicated issue, one that we discuss in greater depth in the Extended Discussion, Appendix S10. In the manuscript text, we focus on adaptative variants to current climate variation and the diversity thereof, which could be important in adaptation to changing climate, following the development presented on our lines 323-326.

Close inspection of Figure 1d in Dauphin et al. 2020 indicates that adaptive loci should show lower diversity in core populations (high "habitat suitability") than in marginal populations (low "habitat suitability") when migration < selection, i.e. when there is some level of population isolation. In contrast, when migration > selection, core populations are expected to have greater diversity at both neutral and adaptive loci (Fig 1b, d). In their Figure 5d, we see that their Pinus populations in

areas of greater habitat suitability (core niche conditions) have lower expected heterozygosity (an indicator of genetic diversity) at putatively adaptive loci than do marginal populations. This result supports the use of climate niche marginality to indicate the populations in which one may find high diversity of variants that are important to climate adaptation and, potentially, response to changing climate. We have drawn this connection, supported by citation of seven other publications (please see lines 323-326). We support the relationship between relative niche marginality and functional variants involved in adaptation to climate with citation to Dauphin et al. 2020 and Perez-Navarro et al 2022 (please see lines 624-627). And we acknowledge that these important variants may be lost due to climate change (please see lines 326-330). We tie this risk to a call for the development of global genetic monitoring frameworks to anticipate climate impacts, by calling for monitoring across climate gradients (please see lines 330-333).

5. Marginality was calculated based on 19 bioclimatic variables. If a site belongs to the 25% most climatically marginal sites, it was defined as marginal. This means that there are marginal habitats for both low and high temperatures. However, for instance, low temperatures will not carry any adaptive diversity for future climate change. The models and assumptions used in the analysis should be more specific and refined.

Response: We thank the referee for the opportunity to clarify this issue.

We first note that the reviewer seems to be critiquing the conservation of genetic diversity in cold marginal (leading edge) populations - the reviewer seems to emphasize that for example 'cold adapted' populations have no utility under future climate change. We disagree with this view of the genetic diversity needed under climate change. Scientists do not (and to some degree cannot) know the full extent of climate change. However, it is known from climate models at global and regional scale that climate change is not simply warming temperatures. Different regions will be impacted differently, with some regions in fact receiving more extreme colds and also 'late frosts' which can damage organisms not adapted to them. Species are also predicted to move in different directions, and not always due to warmth. For instance, climate-driven changes in precipitation or hydrological patterns can strongly affect species' movement patterns. Lastly, the traits vital for cold adaptation (in plants), i.e., affecting desiccation tolerance, water use efficiency, short growing season, can also be helpful in heat and drought conditions. We feel that the reviewer's assumption that the only alleles needed in the future would be for 'heat adaptation' is an over-simplified view of climate change impacts. Conserving marginal populations in all conditions will be valuable for species' persistence and adaptation.

Based on our reasoning, we initially believed it would be sufficient to present the development in reference to all niche margins, and include maps of the future distribution of total marginality for comparison. We agree with the referee in that populations in cold marginal conditions are unlikely to hold at high frequency gene variants adapted to high temperature per se. We also agree that the warm-marginal (i.e. trailing edge) populations are most relevant to the development presented in the manuscript, and to the conservation of genetic diversity that is in danger from climate change. We now clarify that we identify and distinguish the dichotomy in marginal habitat, in that we provide an additional figure and use the Swiss stone pine as an example. Please see the new Figure 4. In response to the referee's comment, we also have entirely changed the presentation of species-specific information, available comprehensively in Appendices 3-6, in Supplementary Materials. Instead of three maps (current, future, and change), we now present two. The first map presents areas within the current range that correspond to core, leading edge marginality and trailing edge marginality. A second map of change in conditions between the current and future periods indicates the future distribution of core and marginal conditions. Throughout the manuscript, the focus is now on trailing-edge marginality.

6. Moreover, the researchers utilized the most extreme climate model (SSP5-8.5) to forecast the forthcoming alteration in marginality without providing any rationale and without conducting a comparison with other models.

Response: We agree that we did not provide sufficient detail in justifying our choice. Our rationale for choosing the most extreme climate scenario is that carbon emissions have not tended toward more moderate scenarios and climate change is currently in line with IPCC worst-case scenarios. Further, the International Energy Agencies and others indicate that carbon emissions in 2022 (Liu et al 2023) continued to grow globally post-Covid19. Nonetheless, we recognize the importance of acknowledging other scenarios, despite their potentially excessive optimism. We now include species-level maps for changing marginality that are in line with a more-moderate scenario, SSP 3-7.0. We also elaborate on justification for our focus on the SSP5-8 scenario and associated climate projections for the future (please see lines 891-892). We present graphical materials that use all six combinations of GCMs and SSPs for all species maps (Appendices S3-6, Supplementary materials) and for all country-level summaries and regional patterns at the pixel level (Appendices S7-9, Supplementary materials). In almost all cases, trends vary somewhat among these scenarios, but the general conclusions are not affected.

7. Furthermore, it was not specified which of the five models available for the time span 2051-70 at https://envicloud.wsl.ch/#/?prefix=chelsa/chelsa_V2/GLOBAL/climatologies/ was selected, nor was the rationale for the choice provided. It should be noted that the different models exhibit varying levels of climate sensitivity, and therefore the selection of a particular model is significant. This approach may raise questions about the robustness and generalizability of their finding

Response: We believe that the referee is correct and that by choosing only one model, we may have biased the results in some unknown way, without acknowledging that there is variation among future climate simulations within an SSP. We now provide results for marginality distributions that are generated with climate data from three global circulation models (GCMs), IPSI, UKISM1-0-LL, and MPI and two SSPs. We chose these six model combinations because they vary substantially in average effects of global warming. Please see lines 891-895.

8. The manuscript completely ignores insects, which are currently suffering a significant die-off. Hence, the selection of species for the marginality analysis appears arbitrary, and many relevant species from different ecosystems are missing.

Response: We thank the referee for drawing attention to this issue, and we fully agree that other groups than those considered here should be included in future monitoring of genetic diversity. However, here we wanted to focus on proposing a new approach to population genetic monitoring that addresses species niches and climate change. This general approach can be applied to any species in any taxonomic or functional group, or ad hoc group, provided the data are available and of sufficient quality. For this reason we used species in groups for which we considered IUCN range maps reliable, and species for which we believed species occurrence data were well reported. We realize it was not clear enough in the submitted manuscript, and we have now clarified the species selection in the revised manuscript. Please see lines 852-853. Insect species appear in our data on monitoring projects and we now include a map of insect monitoring by country in Appendix S1, Supplementary Materials.

19

--Title--

9. As mentioned above the term ‘capacity’ is very misleading in the context reported in the manuscript

ess
is

Response: We refer the referee and editor to our response to this referee's Comment 1.

--Abstract--

10. >>Line 195ff and key words: The term 'genetic monitoring' encompasses monitoring programs that utilize environmental DNA (eDNA) approaches. However, these programs are primarily focused on inter-species diversity, rather than monitoring intra-species genetic diversity. Given the rapid development of various fields, it is important to ensure the specificity of definitions.

Response: We are open to the possibility that we did not completely justify our choice of a definition of the monitoring of population genetic diversity in the abstract. This was likely due to the limited length of the abstract and, thus, our inability to include much highly relevant information. We point the referee to the text where we clearly adopt the Schwartz et al. 2007 definition of Category II monitoring, because we specifically exclude the use of genetic methods for the detection of hard-to-detect species, species in harvest industries and other uses of genetic tools for species identification (Category I monitoring, Schwartz et al. 2007). We have clearly defined monitoring of population genetic diversity by citation in the main text. Please see new line numbers 372-378. We add the word 'population' to the key words, line 299.

--Introduction--

11. >> Line 222ff: I disagree with the general statement that populations in extreme climatic conditions, such as those near their climatic niche margins, are particularly relevant to species potential for adaptation to changing climate. Such marginal populations usually have lower genetic diversity and are adapted to specific environments (Dauphin et al. 2020), making them highly vulnerable to any environmental change. I respectfully disagree with the broad assertion that populations in extreme climatic conditions, such as those near their climatic niche margins, are particularly relevant to species potential for adaptation to changing climate. Typically, populations situated at the edge of their ecological ranges tend to exhibit lower genetic diversity and are adapted to specific environmental conditions (Dauphin et al. 2020), because they lost a lot of genetic diversity during the colonization process (Wegmann et al. 2006; Takahashi et al. 2016) and adaptation (Ellegren & Galtier 2016). Consequently, these populations can be particularly susceptible to the impacts of environmental changes.

*Response: The importance of marginal populations as sources of adaptive variation for potential response to climate-forced environmental degradation depends on the relationship between rates of selection and migration, as we describe above in response the referee's similar comment number 4. The Referee has stated the exact reason why we believe populations that (for example) approach the (hot/dry) margins of their climate niche are of interest; because they likely have adaptations to those particular conditions. The genetic variants present in those populations are at risk of being lost when in the future areas with marginal conditions become uninhabitable, i.e., outside the environmental niche. When those adaptations are the result of private alleles in marginal populations, loss of those populations means loss of a source of variation that is likely relevant for the ongoing adaptation of populations that will become increasingly marginal as climate change progresses, including populations in areas of core habitat that become marginal in the future. We and Dauphin et al 2020 are not saying that **all** marginal populations have these adaptations, but that they likely have them, and that trailing-edge marginal populations are likely a source of variants that may increase fitness in relatively hot-dry environments, having been exposed to those environments. The conditions for relatively high genetic diversity at adaptive loci were discussed above in the context of Dauphin et al. 2020 in response to the referee's comment 4. Please see the response to comment 4, above.*

20

ess
is

Notably, while the two empirical papers mentioned by the referee present carefully conducted research and are interesting, neither should be taken as rebutting or contradicting our development of the potential importance of marginal populations for climate change response on lines 319-326. The Wegmann et al. 2006 paper only models neutral diversity and ignores dynamics that might arise when variants have differential fitness across environmental gradients. Dauphin et al. 2020 demonstrate empirically that patterns at neutral loci and at those loci likely under selection can differ greatly across gradients of environmental suitability. The Takahashi et al. paper reports lack of local adaptation due to low genetic diversity in populations at the cold environmental margin. These are likely leading-edge populations, in contrast to the examples we cite and the focus of our study. As we noted in our response to Comment 4, the relationship between genetic variation and ecological marginality is an ongoing area of research, but many studies have found local adaptation across environmental gradients.

12. >> Line 233ff: The authors have stated that populations inhabiting areas near the limits of warm and dry habitats possess significant adaptive genetic diversity. However, their analysis of niche marginality does not specifically target this particular marginal habitat. Instead, it encompasses a broader range of marginal niches on a global scale. Hence, their reasoning and performed analysis are not in line.

Response: In new figures we now distinguish between leading and trailing margins, also known as conditions near cold and hot niche limits, as we described further above.

13. >>Line 247: The utilization of numerous abbreviations within the manuscript impairs its readability, particularly when the abbreviations PDG and GDP are utilized interchangeably, solely differing in the order of their letters. Furthermore, the abbreviation GMC may require alteration since it fails to accurately represent the shown data.

Response: We thank the referee for noticing this. PDG was a typo that seems to have occurred once, and we have changed it to PGD, for Population Genetic Diversity. We hold that GDP is a broadly recognized abbreviation for Gross Domestic Product. Both abbreviations are used in specific contexts, so there should not be a problem in the absence of typos. We change GMC to GME to reflect the change from 'capacity' to 'effort', as requested by the referee. The terms represented by these three abbreviations are central to the conceptual development of the manuscript and the results, and they are repeatedly referred to in many places in the manuscript. We hope that the use of these three abbreviations (without typos) can be considered acceptable, since they do shorten the manuscript and do not constitute excessive abbreviation. We also use COST repeatedly, for the obvious advantage over 'European Commission's Cooperation in Science and Technology program'. Additional abbreviations in the manuscript include EU, GCM, GLM, IUCN, SAC, SSP, and WoS, all of which are explicitly defined, used at least twice (primarily in the Online Methods section), and likely already familiar to a large portion of the potential readership because of their broad use in the relevant technical and popular literature.

--Results--

14. >>Line 278: At present, there exist 41 countries affiliated with COST, as opposed to the here stated 38, indicating that the manuscript is outdated.

21

Response: Ukraine, Georgia and Armenia entered COST as Full Members, 31.03.2022 and 10.11.2022, respectively, after we concluded our data collection on monitoring programs. We correctly reference 38 COST countries and provide the dates of entry to show that the three

ess
: is

countries entered after data collection. See lines 732-735. For sake of clarity, we also state in line 732 that our data collection effort ran until 31 December 2021, so that any reader can easily determine the timespan window used for our study. While we agree that our dataset may miss more recent monitoring efforts, we are convinced that the outcome of our analyses still adequately reflects the current situation. Please see our response to this referee's comment two (2).

15. >> Line 296: The COST specification indicating that the results are specific to COST countries may give the impression to the reader that the manuscript is intended solely for COST countries and may have limited relevance to a wider audience.

Response: We acknowledge that the results are specific to the defined study area, which consisted of COST full member countries as of the fall of 2019 and spanned much of the European continent. The implications of the results for the importance of conducting genetic samples across entire environmental gradients is, however, generally applicable to any taxon for which genetic monitoring can be justified, and to any region of Earth. We have clarified this on lines 660-663. Our message of the importance of establishing monitoring across climatic gradients is explicitly extended to areas outside of COST countries on lines 660-663.

16. >>Line 351: Assuming that species' climate niche remains stable over time is a strong assumption, and furthermore, it has been demonstrated that species distribution models can vary even within species (Razgour et al. 2019; Chardon et al. 2020)

Response: We agree that the assumption of niche stability is relevant for all studies that present projections of the niche, whether modeled or analyzed statistically, to a different temporal period and corresponding estimated climate. We transparently acknowledge this assumption (lines 905-907), and due to space limitations and the large literature on this topic (see reviews cited on line 907), we believe that refraining from expounding on the validity of the assumption is appropriate here. Regarding niche variation within species, we also state that we calculate a niche estimate across the species' entire range, thus being an aggregate estimate across all populations. Please see lines 854-857, where we incorporate this information.

17. >>Line 371: Overall, I am skeptical about the meaningfulness of summarizing niche marginality across different species, as most species have highly specific niches, as is well-known in amphibians (Rodriguez-Rodriguez et al. 2020).

Response: We acknowledge that it is unlikely that two species would have the same climate niche, as estimated here, unless they had the exact same distribution data from GBIF and the same range polygons from the IUCN. However, that is not the point of constructing maps of species joint distribution of niche marginality. We construct species joint niche marginality maps, and maps showing the change thereof, to indicate the geographic distribution of areas where populations of many species are likely near their niche margins, and where many species will in the future newly experience marginal environmental conditions due to climate change. We believe that these areas will be important for conservationists who seek to mitigate climate change impacts for multiple species, and for conservation geneticists who are interested in designing and optimizing field collections that target many taxa. In constructing the joint niche marginality maps, we impose no restriction that species must have similar niches. We expand the text a little to improve our description of the construction of maps of species joint niche marginality. Please see lines 907-910. We use information on species joint niche marginality to direct recommendations on policy related

22

ess
is

to the need for increased monitoring effort of the study groups, demonstrating the meaningfulness and utility of the joint marginality concept. Please see lines 610-614.

--Discussion--

18. >>Line 404: I disagree with the statement that only a small fraction of studies have been missed. In recent years, the research field has rapidly evolved, and many more projects have been initiated since the recording phase (2019-2021). Additionally, the strong focus on the COST countries/project introduced another potential bias.

Response: In response to this referee's comment, we would like to further clarify the issues presented and addressed in response to a similar comment (number 2) by this referee, above. It is clear that the number of monitoring programs can only increase over time, since we counted all programs that met the criterion for Category II monitoring (Schwartz et al 2007). But given the broad representation of researchers contributing to this dataset, it is unlikely that many more studies have been initiated since, in comparison to the number of studies identified by our study. There are two grounds for our claim. First, the new CBD COP 15 Goal A and target 4 were accepted in December 2022, which is recent enough that funding sources will not have fully responded to these developments. Second, while some programs will surely have started in the 7 months since this date, it is difficult to believe that a great number have been initiated in countries with little or no history of monitoring population genetic diversity. If there has been some major response by countries with little history of monitoring PGD, the geneticists from those COST countries and involved as co-authors are unaware of it. We need a suitable baseline on PGD monitoring in Europe in order to evaluate the effects of COP15, and the time span reported here is pre-COP15, making our results a useful baseline for comparison. Additionally, as for many data-gathering/reviewing and data analysis projects in the sciences, we need to set a date to end data collection to be able to perform the data analysis, implying that any study of this kind will be more or less 'outdated', in that there will always be a period between the end of the data collection and the publication of the study, during which other suitable data would have been produced. We refer readers to our response to this Referee's Comment 2.

19. >>Line 511: The author claims that 'The present study suggests that populations towards the warm/dry, retreating niche margins are geographically clustered in Europe.' However, given the presented results, I cannot follow this argument as marginality was not calculated for dry/warm niches.

Response: We have modified the maps of marginality for each species to indicate both the areas of leading (i.e. cold, getting warmer) and trailing (i.e. hot and getting hotter) environments. We now indicate areas that are newly marginal towards the trailing (warm/dry) margin of acceptable climatic conditions, areas of hot and cold marginal areas that remain that way, and areas of previously core climate that become warm marginal in the future. This is shown in a new figure (Fig. 4) that uses the Swiss stone pine as an example, and is stated in words on lines 456-457. We produce new maps of joint niche marginality, focusing exclusively on the trailing edge (Figs. 5, 6 and all additional figures in which marginality is presented). The Introduction and the rest of the manuscript has been modified to focus on trailing-edge niche marginality.

23

--Materials and Methods--

20. >>Line 703: It is unclear why the most extreme climate model (SSP5-8.5) was used to forecast forthcoming alterations in marginality, and additionally it was not specified which of the five models available for the time span 2051-70 at

ess
is

https://envicloud.wsl.ch/#/?prefix=chelsa/chelsa_V2/GLOBAL/climatologies/ was selected. It should be noted that the different models exhibit varying levels of climate sensitivity, and therefore the selection of a particular model is significant. This approach may raise questions about the robustness and generalizability of the findings.

Response: We addressed a very similar comment above and refer the editor and referee to that response, above (see our answer to reviewer 3, comment 6).

-- Figures--

21. >>Figure 6: As the shown "capacity" does not reflect the actual potential, it is unclear what this figure can tell us.

Response: We have described just above the changes we have made. Please see the new figure 7.

21. >>Extended Data Figure 5: The legend at the top of the figure and the text legend do not match, making it difficult to understand the content of the figure.

Response: We now focus exclusively on trailing-edge marginality and have re-constructed figures of species joint niche marginality at both the country level and the pixel level. In both Extended Data Figures 5 and 6, the edited text now accurately describes the figures, which focus on newly arising marginality and loss of suitable conditions, respectively.

Literature

Freyhof, J. and Brooks, E. 2011. *European Red List of Freshwater Fishes*. Publications Office of the European Union. Luxembourg. DOI: <https://doi.org/10.2779/85903>

Myers et al 2000, Biodiversity hotspots for conservation priorities. *Nature* 403:853-858

Neubert, E., Seddon, E., Allen, M.B. and Backeljau, T., 2019. *European Red List of Terrestrial Molluscs*. IUCN: Cambridge, UK and Brussels, Belgium

Schwartz, M. K., Luikart, G. & Waples, R. S. Genetic monitoring as a promising tool for conservation and management. *Trends in Ecology & Evolution* **22**, 25-33 (2007)

Decision Letter, first revision:

11th September 2023

Dear Dr. Pearman,

Thank you for submitting your revised manuscript "Conserving genetic diversity during climate change: Niche marginality and discrepant monitoring effort in Europe" (NATECOLEVOL-23030537A). It has now been seen again by Referee #1, whose comments are below. Unfortunately, Referee #3 was unavailable to re-review the manuscript, and so Referee #1 commented on the response to their earlier comments on their behalf, and found them satisfactory. As you know, Referee #2 only had relatively minor comments in the previous round, and so was not required to re-review. As such, the remaining reviewers find that the paper has improved in revision, and therefore we'll be happy in principle to publish it in Nature Ecology & Evolution, pending minor revisions to satisfy the reviewers' final requests and to comply with our editorial and formatting guidelines.

[REDACTED]

Reviewer #1 (Remarks to the Author):

The authors did a great job addressing my comments, and those of the other reviewers. Switching the focus to the trailing niche limits made the narrative much clearer. I am happy to recommend this manuscript for publication. I had only two minor comments:

Fig 4 is a great addition, except grey contour lines of (a) (d) and (e) are overlapping and going underneath (b) or on top (c) of PCAs, so it is hard to see where one map ends and another one begins. Please define NMI.

L400- remove semicolon

25Our ref: NATECOLEVOL-23030537A

20th September 2023

Dear Dr. Pearman,

Thank you for your patience as we've prepared the guidelines for final submission of your Nature Ecology & Evolution manuscript, "Conserving genetic diversity during climate change: Niche marginality and discrepant monitoring effort in Europe" (NATECOLEVOL-23030537A). Please carefully follow the step-by-step instructions provided in the attached file, and add a response in each row of the table to indicate the changes that you have made. Please also check and comment on any additional marked-up edits we have proposed within the text. Ensuring that each point is addressed will help to ensure that your revised manuscript can be swiftly handed over to our production team.

****We would like to start working on your revised paper, with all of the requested files and forms, as soon as possible (preferably within two weeks). Please get in contact with us immediately if you anticipate it taking more than two weeks to submit these revised files.****

In recognition of the time and expertise our reviewers provide to Nature Ecology & Evolution's editorial process, we would like to formally acknowledge their contribution to the external peer review of your manuscript entitled "Conserving genetic diversity during climate change: Niche marginality and discrepant monitoring effort in Europe". For those reviewers who give their assent, we will be publishing their names alongside the published article.

Nature Ecology & Evolution offers a Transparent Peer Review option for new original research manuscripts submitted after December 1st, 2019. As part of this initiative, we encourage our authors to support increased transparency into the peer review process by agreeing to have the reviewer comments, author rebuttal letters, and editorial decision letters published as a Supplementary item. When you submit your final files please clearly state in your cover letter whether or not you would like to participate in this initiative. Please note that failure to state your preference will result in delays in accepting your manuscript for publication.

Cover suggestions

26We welcome submissions of artwork for consideration for our cover. For more information, please see our [guide for cover artwork](https://www.nature.com/documents/Nature_covers_author_guide.pdf).

Nature Ecology & Evolution has now transitioned to a unified Rights Collection system which will allow our Author Services team to quickly and easily collect the rights and permissions required to publish your work. Approximately 10 days after your paper is formally accepted, you will receive an email in providing you with a link to complete the grant of rights. If your paper is eligible for Open Access, our Author Services team will also be in touch regarding any additional information that may be required to arrange payment for your article.

Please note that *Nature Ecology & Evolution* is a Transformative Journal (TJ). Authors may publish their research with us through the traditional subscription access route or make their paper immediately open access through payment of an article-processing charge (APC). Authors will not be required to make a final decision about access to their article until it has been accepted. [Find out more about Transformative Journals](https://www.springernature.com/gp/open-research/transformative-journals)

Authors may need to take specific actions to achieve [compliance with funder and institutional open access mandates](https://www.springernature.com/gp/open-research/funding/policy-compliance-faqs). If your research is supported by a funder that requires immediate open access (e.g. according to [Plan S principles](https://www.springernature.com/gp/open-research/plan-s-compliance)) then you should select the gold OA route, and we will direct you to the compliant route where possible. For authors selecting the subscription publication route, the journal's standard licensing terms will need to be accepted, including <https://www.nature.com/nature-portfolio/editorial-policies/self-archiving-and-license-to-publish>. Those licensing terms will supersede any other terms that the author or any third party may assert apply to any version of the manuscript.

27[REDACTED]

[REDACTED]

Reviewer #1:

Remarks to the Author:

The authors did a great job addressing my comments, and those of the other reviewers. Switching the focus to the trailing niche limits made the narrative much clearer. I am happy to recommend this manuscript for publication. I had only two minor comments:

Fig 4 is a great addition, except grey contour lines of (a) (d) and (e) are overlapping and going underneath (b) or on top (c) of PCAs, so it is hard to see where one map ends and another one begins. Please define NMI.

L400- remove semicolon

Author Rebuttal, first revision:

Reviewer #1 (Remarks to the Author):

The authors did a great job addressing my comments, and those of the other reviewers. Switching the focus to the trailing niche limits made the narrative much clearer. I am happy to recommend this manuscript for publication. I had only two minor comments:

Fig 4 is a great addition, except grey contour lines of (a) (d) and (e) are overlapping and going underneath (b) or on top (c) of PCAs, so it is hard to see where one map ends and another one begins.

We have improved the figure, now Figure 3, to make the panels more distinct and eliminate lines that extended beyond the panels. Please see the figure.

Please define NMI.

NMI is now defined in the legend of Figure 3, panel 'a', as follows:(a) Current marginal and core areas. Marginal areas are split into trailing and leading edge based on differences between current and future Niche Margin Index (NMI) values (leading edge: positive NMI change, trailing edge: negative NMI change).

L400- remove semicolon

The semicolon has been removed.

Final Decision Letter:

Dear Professor Pearman,

We are pleased to inform you that your Article entitled "Monitoring species genetic diversity in Europe varies greatly and overlooks potential climate change impacts", has now been accepted for publication in Nature Ecology & Evolution.

Over the next few weeks, your paper will be copyedited to ensure that it conforms to Nature Ecology and Evolution style. Once your paper is typeset, you will receive an email with a link to choose the appropriate publishing options for your paper and our Author Services team will be in touch regarding any additional information that may be required

Due to the importance of these deadlines, we ask you please us know now whether you will be difficult to contact over the next month. If this is the case, we ask you provide us with the contact information (email, phone and fax) of someone who will be able to check the proofs on your behalf, and who will be available to address any last-minute problems . Once your paper has been scheduled for online publication, the Nature press office will be in touch to confirm the details.

Acceptance of your manuscript is conditional on all authors' agreement with our publication policies (see www.nature.com/authors/policies/index.html). In particular your manuscript must not be published elsewhere and there must be no announcement of the work to any media outlet until the publication date (the day on which it is uploaded onto our web site).

Please note that *Nature Ecology & Evolution* is a Transformative Journal (TJ). Authors may publish their research with us through the traditional subscription access route or make their paper immediately open access through payment of an article-processing charge (APC). Authors will not be required to make a final decision about access to their article until it has been accepted.  Find out more about Transformative Journals

Authors may need to take specific actions to achieve compliance with funder and institutional open access mandates. If your research is supported by a funder that requires immediate open access (e.g. according to Plan S principles) then you should select the gold OA route, and we will direct you to the compliant route where possible. For authors selecting the subscription publication route, the journal's standard licensing terms will need to be accepted, including https://www.nature.com/nature-portfolio/editorial-policies/self-archiving-and-license-to-publish. Those licensing terms will supersede any other terms that the author or any third party may assert apply to any version of the manuscript.

An online order form for reprints of your paper is available at https://www.nature.com/reprints/author-reprints.html. All co-authors, authors' institutions and authors' funding agencies can order reprints using the form appropriate to their geographical region.

We welcome the submission of potential cover material (including a short caption of around 40 words) related to your manuscript; suggestions should be sent to Nature Ecology & Evolution as electronic files (the image should be 300 dpi at 210 x 297 mm in either TIFF or JPEG format). Please note that such pictures should be selected more for their aesthetic appeal than for their scientific content, and that colour images work better than black and white or grayscale images. Please do not try to design a cover with the Nature Ecology & Evolution logo etc., and please do not submit composites of images related to your work. I am sure you will understand that we cannot make any promise as to whether any of your suggestions might be selected for the cover of the journal.

nature portfolio

You can generate the link yourself when you receive your article DOI by entering it here: <http://authors.springernature.com/share>.

Yours sincerely,

[REDACTED]

P.S. Click on the following link if you would like to recommend Nature Ecology & Evolution to your librarian <http://www.nature.com/subscriptions/recommend.html#forms>

** Visit the Springer Nature Editorial and Publishing website at http://editorial-jobs.springernature.com?utm_source=ejp_NEcoE_email&utm_medium=ejp_NEcoE_email&utm_campaign=ejp_NEcoE for more information about our career opportunities. If you have any questions please click [here](mailto:editorial.publishing.jobs@springernature.com). **